

# Sinkholes, stream channels and base-level fall: a 50-year record of spatio-temporal development on the eastern shore of the Dead Sea

Robert A. Watson[1], Eoghan P. Holohan[1], Djamil Al-Halbouni[2], Leila Saberi[3], Ali Sawarieh[4], Damien Closson[5], Hussam Alrshdan[4], Najib Abou Karaki[6*], Christian Siebert[7], Thomas R. Walter[2], & Torsten Dahm[2]

[1] UCD School of Earth Sciences, University College Dublin, Dublin, Ireland
[2] Helmholtz Centre Potsdam (GFZ), Section 2.1, Potsdam, Germany
[3] University of Minnesota, Department of Earth Sciences, Minneapolis, USA
[4] Ministry of Energy & Mineral Resources, Amman, Jordan
[5] Geographic Information Management, Brussels, Belgium
[6] Department of Environmental and Applied Geology, University of Jordan, Amman, 11942, Jordan
*On sabbatical leave at the Environmental Engineering Department, Al-Hussein bin Talal University, Ma'an-Jordan.
[7] Helmholtz Centre for Environmental Research – UFZ, T. Lieser Str. 4, Halle 06120, Germany

*Correspondence to*: Eoghan P. Holohan (eoghan.holohan@ucd.ie)

**Abstract.** The fall of hydrological base-level is long established as a driver of geomorphological change in both fluvial and karst systems, but few natural occurrences occur on timescales suitable for direct observation. Here we document the spatiotemporal development of fluvial and karstic landforms along the eastern coast of the hypersaline Dead Sea (at Ghor al-Haditha, Jordan) during a 50-year period of regional base-level decline from 1967 to 2017. Combining remote sensing data with close-range photogrammetric surveys, we show that the 35 m base-level fall has caused shoreline retreat of up to 2.5 km, and resulted in: (1) incision of new meandering or straight/braided stream channels and (2) formation of >1100 sinkholes and several salt-karst uvalas. Both alluvial incision and karst-related subsidence represent significant hazards to local infrastructure. The development of groundwater-fed meandering stream channels is in places interlinked with that of the sinkholes and uvalas. Moreover, active areas of channel incision and sinkhole development both migrate seaward in time, broadly in tandem with shoreline retreat. Regarding theoretical effects of base-level fall, our observations show some deviations from those predicted for channel geometry, but are remarkably consistent with those for groundwater-




related salt karstification. Our results present, for the first time in the Dead Sea region, the dual response of surface and subsurface hydrological systems to base level drop as indicated by fluvial and karst geomorphological analysis.

## 1    Introduction

The concept of hydrological base-level is over 100 years old (Davis, 1902), and it is key to understanding changes of the Earth's surface due to tectonic deformation or climate change (Allen, 2008; Best and Ashworth, 1997; Whittaker, 2012). The exact definition of base-level varies between disciplines, however. In fluvial geomorphology, base level is defined as the "imaginary horizontal level to which sub-aerial erosion proceeds" (Schumm, 1993), generally regarded as sea-level. In karst geomorphology, base-

level is defined as "the point of groundwater outflow from the subsurface karst drainage system" (Bakalowicz, 2005).

Regardless of definition, the rise or fall of hydrological base level is long known to result in complex geomorphological responses. Changes in fluvial channel morphology have been proposed to depend on

the rate and magnitude of base level fall, the local relief and hydrological input, and the character of the channel substrate (Leopold and Bull, 1979; Schumm, 1993; Whittaker and Boulton, 2012). In a karst system, base level fall (i.e. decline in the level of the phreatic zone) is envisaged to cause new conduit development (by vadose incision and/or phreatic passage formation at the newly defined base level) and to generate new outflow points (Bakalowicz, 2005; Farrant and Simms, 2011; Ford and Williams, 2007).

Subsequent karstic development may be expressed at the surface by new or accelerated formation of *dolines,* or sinkholes (as we shall refer to them) (Gutiérrez et al., 2014).

The hypersaline Dead Sea represents a regional hydrological base level that has fallen, largely because of anthropogenic-forcing, at a gradually increasing rate since the late 1960s (Lensky et al., 2005). The base

level fell at a rate of 0.5 m yr$^{-1}$ in the 1970's, and at a rate of 1.1 m yr$^{-1}$ in the last decade. In absolute terms, the lake level has declined by 37 m as of 2017 and is forecast to drop a further 25-70 m by 2100 (Asmar and Ergenzinger, 2002; Yechieli and Gavrieli, 1998). The margins of the Dead Sea are undergoing





dramatic geomorphological changes including enhanced stream and river channel incision (Bowman et al., 2010; Moshe et al., 2008; Vachtman and Laronne, 2013), slope instability and landslides (Closson et
al., 2010), as well as the development of several thousands of sinkholes by karstification of salt-rich deposits underlying the lake margins (Abelson et al., 2017; Yechieli et al., 2006). These changes represent substantial geohazards in the Dead Sea region. They have already destroyed or damaged tourism facilities, factories, evaporation pond dykes, highways, link roads, houses and farmland.

While geomorphological changes have been documented in some detail on the western side of the Dead Sea, less information is available about such changes on the eastern side. In this paper, we provide a first detailed documentation of the geomorphological evolution of the main sinkhole-affected site on the eastern shore of the Dead Sea, at Ghor al-Haditha in Jordan, over the 50-year period from the start of the base level drop in 1967 to 2017. Our aims are to discern factors controlling the evolution of the new
landforms, and to examine how hydrological or karstic aspects interact in the context of the base-level fall.

## 2   Tectonic setting and geological framework

The Dead Sea is the hyper-saline terminal lake of the Jordan River (**Figure 1A**), and it lies within the ~150 km long and ~ 8 - 10 km wide Dead Sea basin (Garfunkel and Ben-Avraham, 1996). The basin lies
at a left step (or bend) along the left-lateral Dead Sea Transform fault system. Maximum tectonic subsidence is ~8.5 km around the Lisan peninsula (Ten Brink and Flores, 2012). The basin has hosted several palaeo-lakes of varying size and duration (Bartov et al., 2002; Torfstein et al., 2009). A high-stand of -162 m elevation (with respect to modern global mean sea level, the convention used hereafter) was reached at around 25 ka ago, during the 'Lisan Lake' episode, and the modern Dead Sea initiated after a
major low-stand at around 10 ka ago (Bartov et al., 2002). With the lake's decline from -395 m (1967) to -431 m (2017) it has divided into northern and southern parts; the latter is now taken over by industrial salt evaporation ponds.

 

The Ghor al-Haditha study area, which is about 25 km$^2$ in size, is situated on the southeast shore of the
northern Dead Sea (**Figure 1A**). The area lies in a zone of tectonic complexity at the eastern basin margin,
where subsidence is relayed between several major tectonic structures along a ~ 15° bend in the Dead Sea
Transform system. The major, left-lateral, N24°-trending Wadi Araba fault terminates a few kilometres
south of the area, further north of which basin subsidence is accommodated by combination of the N0°-
trending Ghor Safi fault and the Ed-Dhira monoclinal flexure. The Ghor Safi fault also forms the eastern
boundary of the actively rising Lisan salt diapir (Al-Zoubi and Ten Brink, 2001; Fiaschi et al., 2017). The
Ed-Dhira monocline terminates against the right-lateral N80°-trending Siwaqa fault, which also down-
throws to the north. Further north again, a N10°-trending escarpment probably reflects the orientation of
another major basin-bounding fault (Khalil, 1992), although the exact location of the fault trace is unclear.

The geology of the Ghor al-Haditha study area comprises folded and faulted sequences of siliciclastic or
carbonate rocks, which are locally overlain by semi-consolidated to unconsolidated lacustrine or alluvial
deposits (**Figure 2**). Hydrogeologically, there three principal aquifer units: (1) a lower sandstone aquifer
comprising the Ram group and Kurnub formation of Cambrian to early Cretaceous ages, respectively; (2)
an upper carbonate aquifer spanning the Ajlun and Belqa groups of late Cretaceous to early Tertiary age;
and (3) a superficial aquifer in the Lisan formation of Plio-Pleistocene age (Khalil, 1992).

The Lisan formation deposits at Ghor al-Haditha comprise poorly-sorted, semi-consolidated to
unconsolidated sands and gravels interbedded with minor silts and clays. These deposits, together with
similar but unconsolidated deposits of the Ze'elim formation of Holocene age, form an alluvial fan plain
at between -360m and -380m (**Figure 1B, 1C**). Three major *wadi* (dry river valley) systems terminate
within or adjacent to the study area: Wadi Ibn Hammad, Wadi Mutayl and Wadi al Mazra'a (the latter
lies just outside the study area to the southwest). These drain the uplands to the east and southeast.

The Lisan and Ze'elim formations also comprise lacustrine deposits, some of which are exposed on the
former Dead Sea bed. These form a 'mudflat' or 'saltflat' adjacent the Dead Sea shore (**Figure 1C**), and
comprise laminated to thinly bedded layers of marl, clay, salt and silt interbedded with a spatially variable



proportion of thin to thick layers of rock salt (predominantly halite). Similar lacustrine deposits likely extend in under the alluvial fan (Polom et al., 2018).

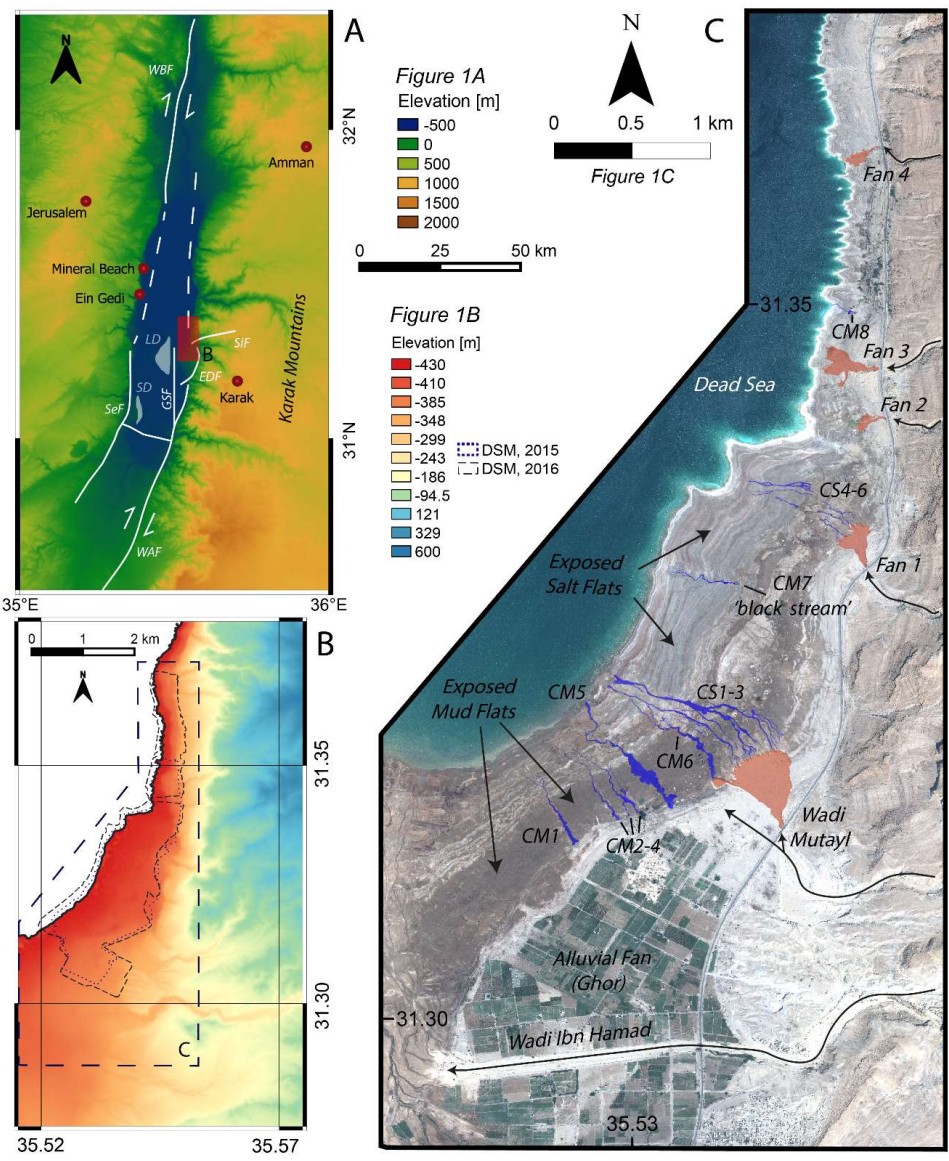

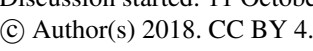


**Figure 1: overview of the Ghor al-Haditha study area. (A) Advanced Land Observing Satellite (ALOS) 30m Digital Surface Model (DSM) of the Dead Sea study area, showing the regional tectonic regime. WBF: Western Boundary Fault; SiF: Siwaqa Fault; LD: Lisan Diapir; EDF: Ed Dhira Flexure; GSF: Ghor Safi Fault; SD: Sedom Diapir; SeF: Sedom Fault; WAF: Wadi Araba Fault. (B) ALOS 30m DSM showing relief in the study area, as highlighted in red in (A), along with the footprints of the 2015 and 2016 drone and field surveys. (C) Pleiades 2017 satellite image of the study area showing main hydrological and geomorphic features referred**

**to later in the study.**

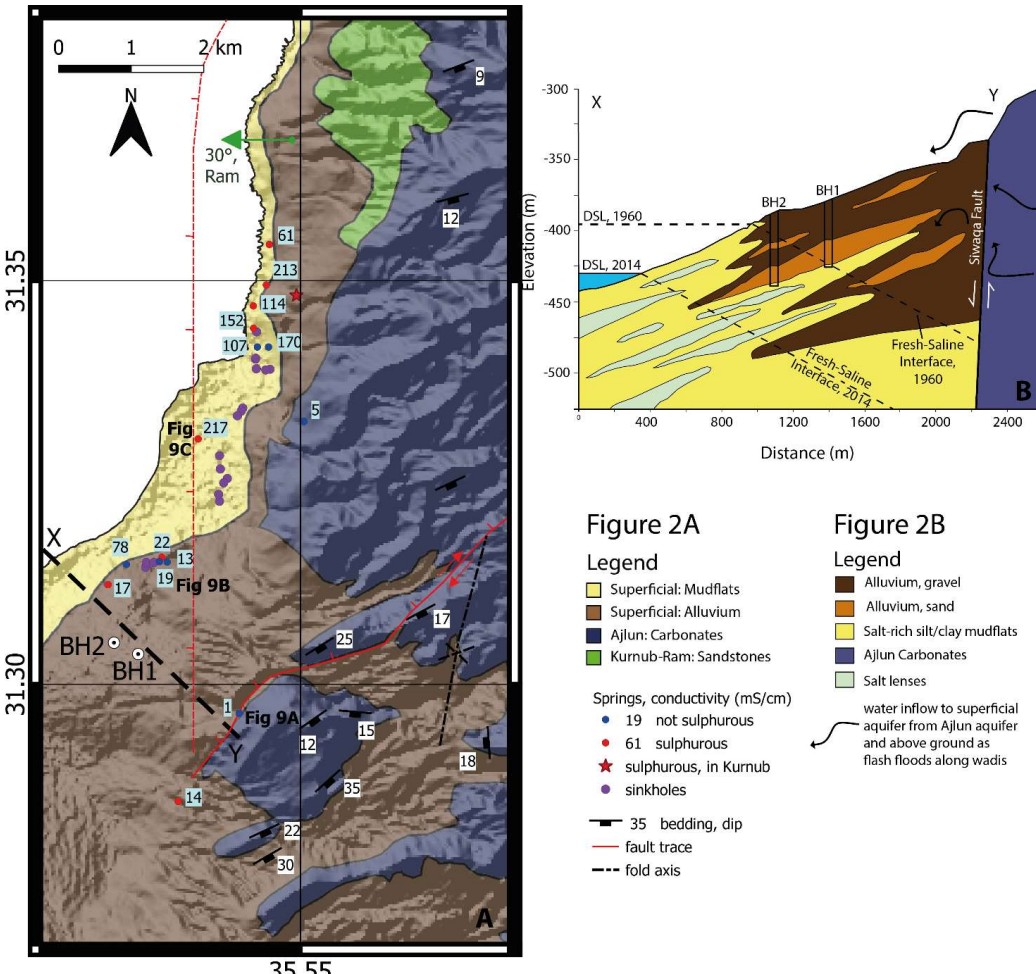

**Figure 2: structural and hydrogeological characteristics of the study area. (A) simplified geological map of the study area, partly based on 1:50,000 scale mapping of Jordanian Ministry of Energy and Mineral Resources (Khalil, 1992) and partly on our own work. The stratigraphy generally dips acutely to the southeast, while striking to the northeast. Also shown is the right-lateral oblique**

**Siwaqa fault, the inferred continuation of the Dead Sea Transform (down-throwing to the east), and the axis of the Haditha syncline.**



The springs and their respective conductivities are from water sampling undertaken by us in 2015, except from the star, which is derived from Khalil (1992). The springs labelled 15A, 15B and 15C are referred to in more detail below. (B) schematic cross-section of sub-surface geology along the black dashed line on the map (X-Y), showing hydrogeological theory predicting the lateral shoreward migration of the interface developed between the hypersaline Dead Sea brine and less saline groundwater ('fresh-saline interface') with time. The two boreholes of El-Isa et al. (1995) are labelled 'BH1' and 'BH2', as on the map. The vertical exaggeration for the cross section is 40.

## 3    Data and Methods

Our data set includes high resolution optical satellite imagery and aerial survey photographs covering the 50-year period from 1967-2017 (**Table 1**). We orthorectified and pansharpened the satellite imagery by using standard algorithms and workflows in the PCI Geomatica software package. For orthorectification of the 2002 – 2010, 2011 – 2013 and 2014 – 2015 satellite imagery, we used the Shuttle Radar Topography Mission (SRTM) Digital Elevation Model (DEM), the Advanced Spaceborne Thermal Emission and Reflection Radiometer (ASTER) DEM, and the Advanced Land Observing Satellite (ALOS) World 3D Digital Surface Model (DSM), respectively. For the Pleiades images from 2016 and 2017, atmospheric correction, orthorectification and georeferencing were conducted by Airbus against the Astrium Elevation 30 global DEM. All pre-2016 images were georeferenced by using nine Ground Control Points (GCPs). Additional co-registration of pre-2016 imagery was performed with respect to the 2017 Pleiades imagery by using tools from the Geospatial Data Abstraction Library (GDAL) with numerous manually-selected tie-points. In the case of the 1967 image, the use of ESRI online World Imagery was also necessary for further co-registration due to the geographical limits of the 2017 Pleiades imagery.

Close-range photogrammetric surveys undertaken in October 2014, October 2015 and December 2016 provide yet higher resolution orthophoto mosaics and DSMs (for survey limits, see **Figure 1B**). The surveyed areas were imaged from a helikite or drone at a height of ~100 m with a 16 Megapixel (MP) Ricoh GR camera (2014), a 12 MP GoPro Hero4 camera with modified lens (2015) or with a 12 MP DJI Phantom 3 inbuilt camera (2016). During each survey, 50-60 temporary GCPs were measured with a Trimble ProXRT differential GPS receiver with real time corrections. Al-Halbouni et al. (2017) detail the procedure for generating these orthophoto mosaics and DSMs. The internal horizontal and vertical uncertainty of the DSMs is estimated to be: 2014 (10 cm, 11 cm), 2015 (12, 17 cm) and 2016 (37, 31 cm).

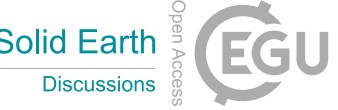



In 2015, we also made a preliminary survey of water sources in the study area. Samples for $\delta^2$H and $\delta^{18}$O were taken unfiltered from springs in bedrock on the landward side of the sinkhole area, from springs within the sinkhole area or on its seaward side, and from ponds within individual sinkholes (**Figure 2A**). Temperature and electrical conductivity were measured in-situ at each sample location with a Hach

HQ40D Portable Multi-meter. $\delta^2$H and $\delta^{18}$O were determined at the UFZ by isotope ratio mass spectrometry (IRMS delta S; Finnigan MAT). For determining $\delta^{18}$O with an analytical precision of ±0.1‰, the standard $H_2O$–$CO_2$ equilibration method (Epstein and Mayeda, 1953) was applied. $\delta^2$H was determined using the chromium technique (Gehre et al., 1996), which does not require corrections for salt effects. Analytical precision is ±0.8‰. All isotope ratios are reported relative to the Vienna Standard

Mean Ocean Water (VSMOW).

All data were integrated and analysed within a Geographic Information System (GIS) software package (Q-GIS). The number and extent of remotely-sensed sinkholes represent minima, as local farmers have filled in sinkholes to mitigate disruption to their work. Therefore, we also include information from

sources that undertook earlier field surveys in communication with local farmers (El-Isa et al., 1995; Sawarieh et al., 2000; Closson and Abou-Karaki, 2009). In addition, we combine our mapping of the coastline through time with historical measurements of the Dead Sea level from the Israel Marine Data Center (ISRAMAR) and the Jordanian Ministry of Water and Irrigation (MWI) to reconstruct the former Dead Sea bathymetry in the study area.




| | Data Source | Aquisition Year(s) | Resolution (m/pix) |
|---|---|---|---|
| **Optical Satellites** | Corona | 1967, 1968, 1970 | 2.0 |
| | Quickbird | 2002, 2004-2007, 2012 | 0.6 |
| | Ikonos | 2006 | 0.8 |
| | Worldview 1 | 2008, 2011, 2012 | 0.5 |
| | GeoEye-1 | 2009-2010 | 0.5 |
| | Worldview 3 | 2014 | 0.3 |
| | Pleiades 1a | 2013, 2015 - 2017 | 0.5 |
| **Aerial Surveys** | RJGC Aerial | 1981, 1992, 2000 | 0.6 |
| | Drone and Helikite surveys | 2014 - 2016 | 0.1 |

**Table 1: Sources and resolution of remote- and near-sensing data used in this study. RJGC = Royal Jordanian Geographic Centre. The spatial resolution of the dataset varies from 1.8 – 0.1 metres per pixel. The temporal resolution of the dataset is decadal from 1970 – 2010, and annual from 2004 – 2017.**

## 4    Results

### 4.1    Base level fall and shoreline retreat

The Dead Sea level drop has resulted in a dramatic retreat of the shoreline in the Ghor al-Haditha area. As of 2017, the shoreline had retreated from its 1967 position by a minimum of 0.3 km in the north to a maximum of 2.5 km in the south. The rate of retreat in the southern part of the study area accelerated from < 10 m/yr between 1967-1980 to an average rate of ~45 m/yr between 2000-2017 (**Figures 3 B, C**). In the north of the area, the rate of retreat has been a steadier of about 7-8 m/yr. **Figure 3D** shows that the rate of shoreline retreat is correlated non-linearly with the former bathymetric slope.




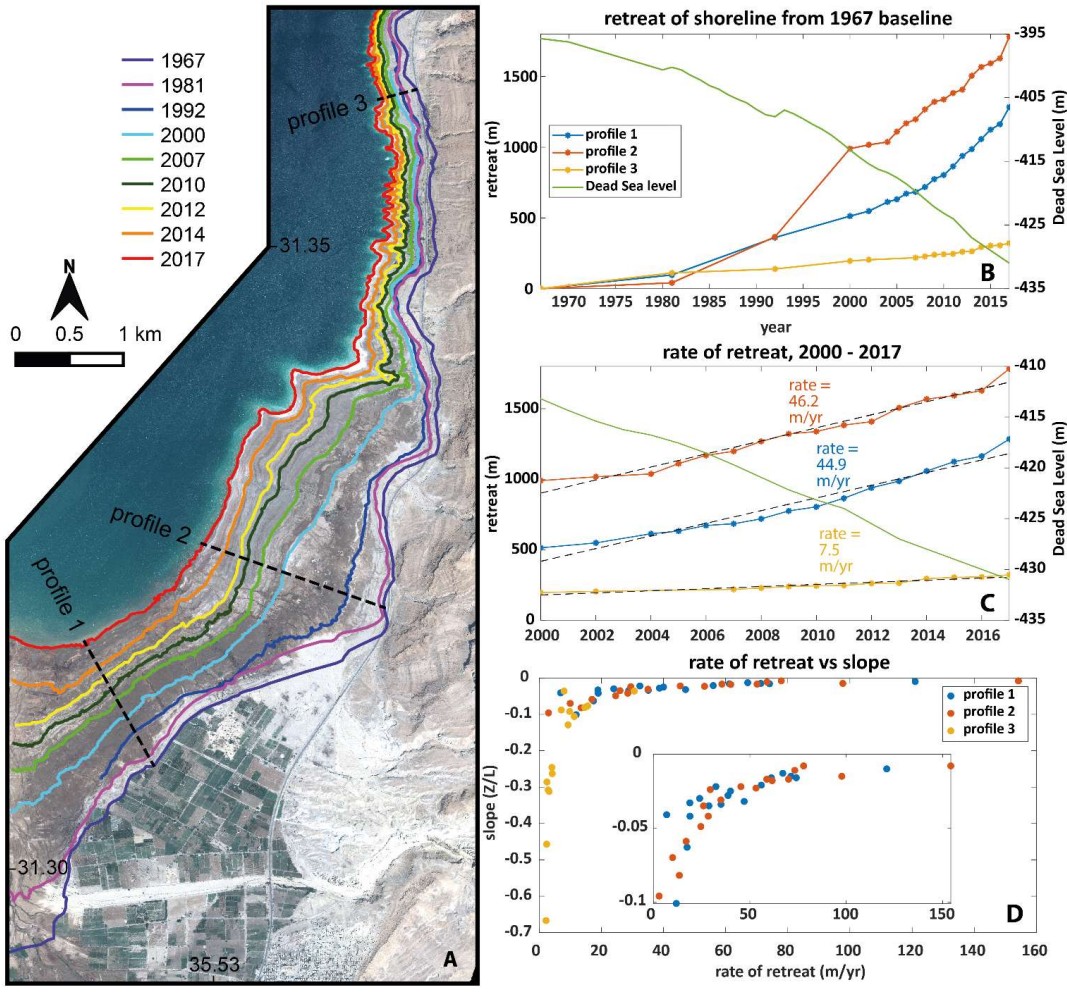

**Figure 3: Dead Sea level fall and shoreline retreat in the Ghor al-Haditha area of Jordan from 1967-2017. (A) Map of shoreline retreat with time, produced from satellite and aerial imagery. Also shown here are plots of the Dead Sea Level and of the shoreline position along profiles in the Ghor al-Haditha area over the periods: (B) 1967-2017 and (C) 2000-2017. (D) Plot of rate of shoreline retreat against bathymetric slope for Profiles 1-3. The inset shows a close-up of the data from Profiles 1-2 for clarity.**





### 4.2 Surface erosion and deposition

#### 4.2.1 Stream channel incision into the former lake bed

Two main stream channel morphologies have been cut into the exposed salty-marl deposits of the former Dead Sea lake bed: meandering (CM) and straight (CS) (**Figure 4**). The heads of all meandering channels have developed at spring points (usually one per channel). Such springs lie either at the alluvium/mudflat boundary or within the mudflat deposits. The heads of straight channels initially developed some distance out on the mudflat, commonly downslope from the terminations of active alluvial fans, and are typically branched.

As the shoreline has retreated, both channel types have grown seaward. While the straight channels also show upstream growth (e.g. CS1-3 in **Figure 4**), most meandering channels show little or no upstream growth (e.g. CM1-4 and CM6 in **Figure 4**). Established sections of both channel types also widen progressively with time. From field observations, channel widening is commonly associated with fault-delimited slumping of the channel sides (Al-Halbouni et al., 2017). These lower sections of the straight channels are commonly braided and contain deposits of sand to cobble clast size. Deposits within the meandering channels are mud to silt size.

An unusual meandering channel is CM5. This formed in 2012 with its head initially located in the middle of the mudflat (**Figure 4**). Channel incision then progressed rapidly upstream over three months towards the alluvium/mud-flat boundary, in association with the drainage of a lake there (see Al-Halbouni et al., 2017 for details). Co-incident with the establishment of CM5, the growth of nearby meandering channels CM1-4 has diminished markedly.

In profile, both meandering and straight channels are 'V-shaped', and they both narrow and shallow seaward along their lengths (**Figure 5**). The straight channels additionally narrow and shallow landward toward the adjacent active alluvial fans. The channel width/depth (W/D) ratio seems independent of the substrate material (salt- or mud-dominated) (**Figure 6A**). All channels in all materials display W/D ratios of 3 – 15 in the channel half nearest the channel head. Straight channels show markedly increased W/D





ratios of 10-40 in their lower sections (**Figure 6A**), however, a variation that is associated with decreasing

slope of the former seabed (**Figure 6B**). In contrast, W/D ratios of meandering channels remain unchanged as one progresses downstream, and are apparently unrelated to slope. Sinuosity of the meandering channels is 1.1-1.7, with a general slight increase along the upper three quarters of the channels, then decreasing to 1.1-1.3 in the lowermost reach (**Figure 6C**). Development of sinuosity in the meandering channels also seems independent of slope (**Figure 6D**).






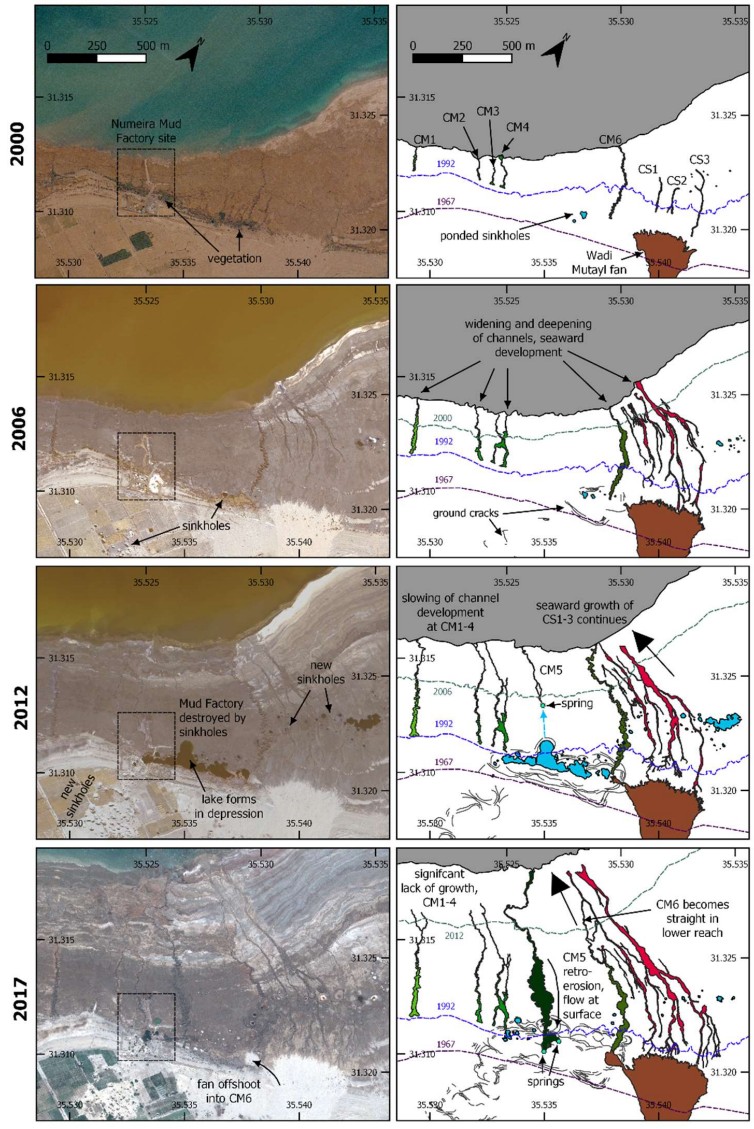

**Figure 4: Evolution of meandering and straight/braided stream channels in the salty mudflat deposits from 2000 –2017. The left column shows aerial or satellite imagery. The right column shows maps of channels (red = straight/braided, green = meandering), the Wadi Mutayl alluvial fan (brown), ground cracks denoting the limits of a large-scale depression, and depression or sinkhole-hosted ponds (blue). Dashed purple, blue and green lines indicate the 1967, 1992 and 2006/2012 shorelines, as labelled.**






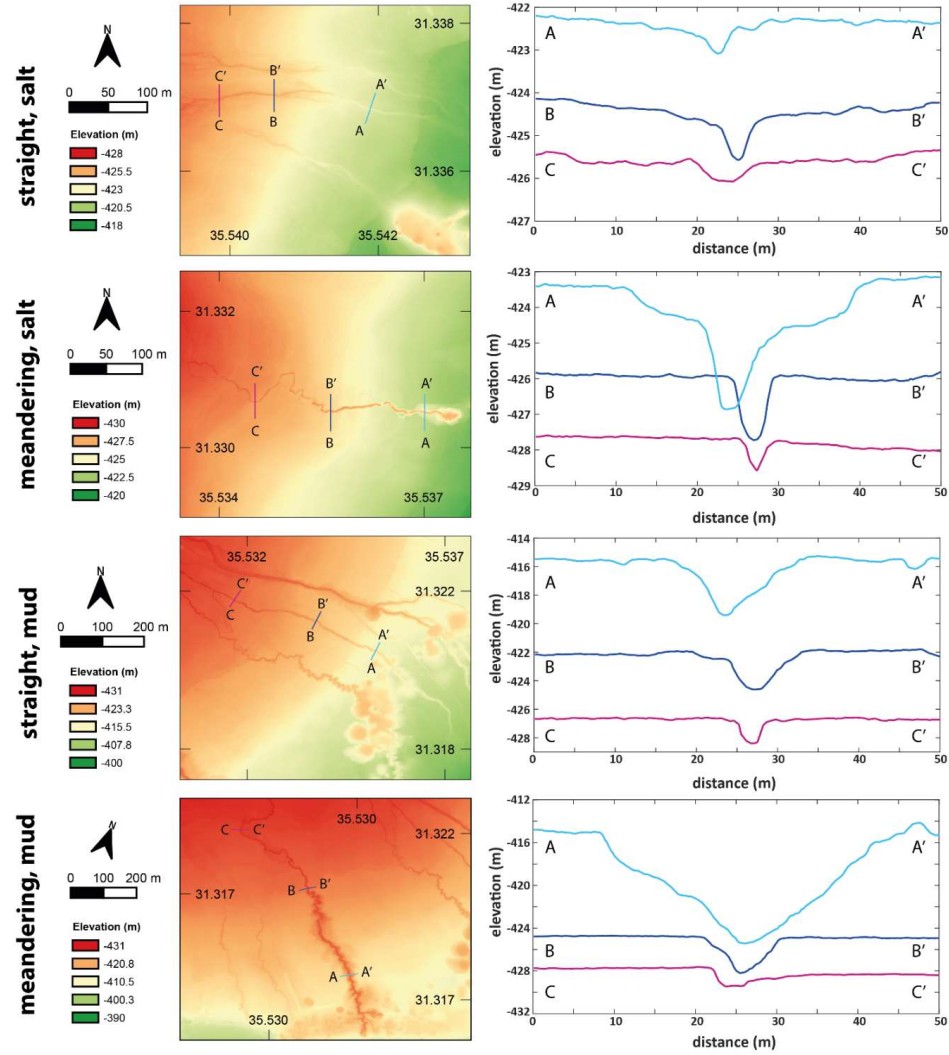

**Figure 5: Representative cross-sectional profiles of stream channels incised into the exposed lacustrine deposits of the former lake bed. Left column shows the 2016 photogrammetric DSM with profile locations; Right column shows the channel profiles.**






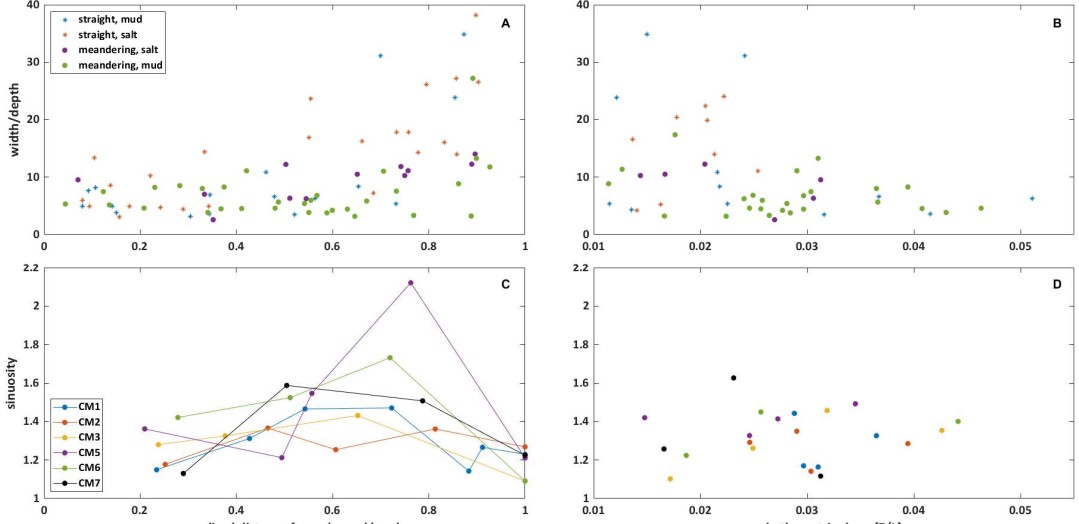

**Figure 6: Quantitative analysis of channel geomorphology. (A) Plot of W/D ratio along the lengths of representative meandering and straight stream channels (locations in figure 1). (B) Plot of W/D ratio against former bathymetric slope. (C) Plot of sinuosity along meandering channels (locations in figure 1). Distance is from head to mouth following the maximum valley slope. (D) Plot of sinuosity against former bathymetric slope.**

### 4.2.2  Stream channel incision into the old marginal alluvial fans

Stream channel incision has also occurred into the pre-recession alluvial fan deposits at the former Dead Sea margin (**Figure 7**). Data here are presented for channels related to the new alluvial fans 2 - 4, as they are covered by our close-range photogrammetry surveys, but similar incision is seen in the larger Wadi Mutayl fan also. These channels have W/D ratios of 2 - 6 and, in contrast to the channels in the exposed lake bed, are trapezoid-shaped in cross sectional profile. Deposits within the channel are of coarse sand, gravel and cobbles.



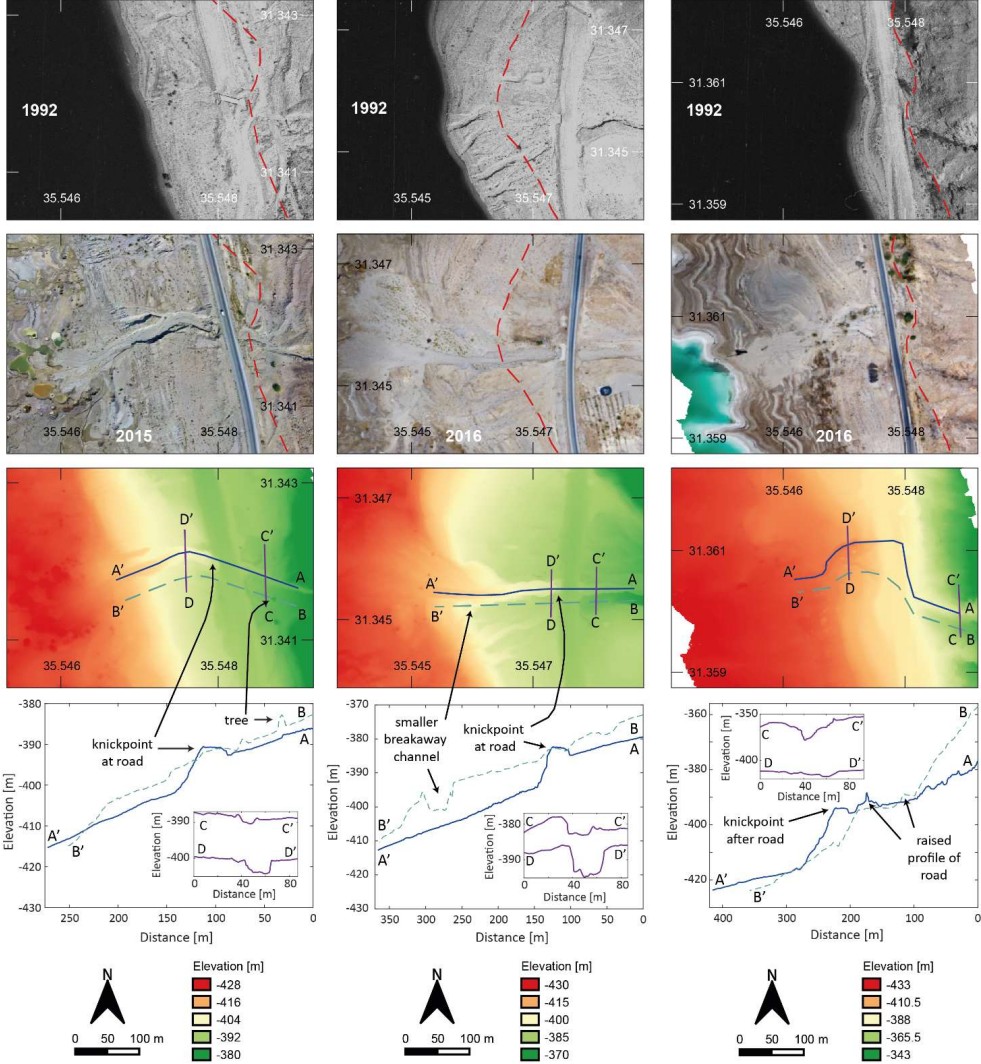


**Figure 7: Alluvial channel incision and fan growth proximal to the Dead Sea highway [Jordan Valley Highway 65]. Top row: aerial imagery from 1992, taken during the road construction. The dashed red line is the 1967 shoreline. Second row: orthophotos from the 2015 (fan 2) and 2016 (fan 3, fan 4) photogrammetric surveys. The channels incise into the old alluvial fan deposits, with fresh deposition of fan material onto the old lake bed. Older alluvial fan deposits appear darker grey in the orthophoto, whereas fresh deposits are lighter coloured. Third row: DSMs derived from respective photogrammetric surveys. Bottom row: topographic profiles along the channels (A-A') and along the non-incised fan adjacent the channel (B-B'). Also shown are profiles immediately upstream (C-C') and downstream (D-D') of the road bridges.**






### 4.2.3 Alluvial fan growth

In addition to fluvial erosion, fluvial deposition has caused alluvial fans to prograde onto the former Dead

Sea bed from adjacent wadis (**Figures 4 and 7**). The absolute growth of these fans in terms of area varies over several orders of magnitude (**Figure 8A**) depending on wadi size. Fan growth at the mouth of the Wadi Ibn Hammad has curtailed by engineering works in the 1980s to restrain and straighten its course down to the shoreline (See **Table 1**). Normalised growth rates from 2002 onward are generally higher in the northern part of the area (**Figure 8B**). This perhaps reflects the belated uncovering of the salt- or mud-

flat there.

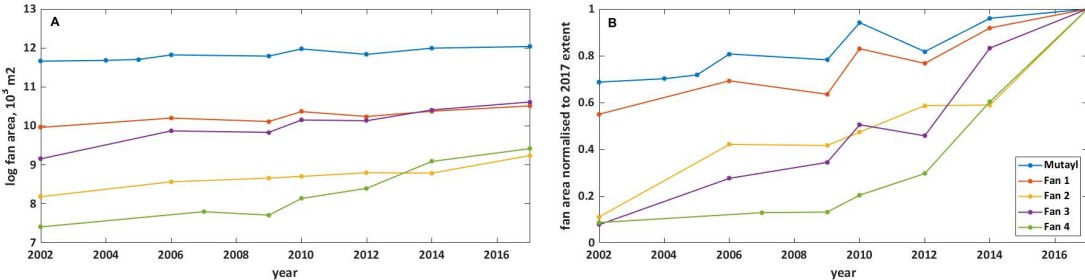

**Figure 8: Alluvial fan growth in the study area from 2002 to 2017, in terms of (A) absolute areas and (B) area normalised to 2017 values. For fan locations, see Figure 1.**




### 4.3 Surface subsidence and collapse

#### 4.3.1 Sinkholes

Sinkhole formation began at Ghor al-Haditha in the mid-1980s in the southern part of the study area
(**Figure 9**). Initiation of new sinkhole development subsequently shifted north-northeast-ward, roughly
parallel to the coastline. The most active area is now adjacent the Dead Sea highway in the northern part
of the study area. In detail the sinkholes have initiated in clusters, with gaps between earlier clusters filled
or reduced as new sinkholes and sinkhole clusters form. After initiation, sinkhole development has
generally migrated seaward to variable extents of between 100 – 500 m.

The growth rate of the sinkhole population at Ghor al-Haditha was exponential between 1985 and 2009,
but it has been roughly linear from 2009-2017 (**Figure 10**). With a notable peak in 2005 also, the
maximum occurred in 2009 with 134 new holes. Since 2009, the rate of sinkhole population growth has
stabilised at ~60 new holes per year. We estimate that at least ~1150 sinkholes have formed in the area
between 1985 and 2017.

The size and morphology of individual sinkholes is linked loosely to the material in which they form
(**Figure 11**). In general, sinkholes have diameters of 1 - 40 m, but some sinkholes in the mud-dominated
lacustrine deposits have diameters of over 70 m (**Figure 11A**). The mode of sinkhole diameter is 4 – 8 m
in 'salt', 4 – 12 m in alluvial sediments, and 8 – 16 m in lacustrine 'mud'. Holes formed in the mud- and
salt-dominated materials have lower depth/diameter (De/Di) ratios than holes formed in alluvial
sediments (**Figure 11B**). Alluvium-hosted holes show the least variance from the linear regression model
calculated; mud-hosted holes are highly variable in their De/Di properties. Regardless of materials,
eccentricity of sinkhole circumferences is usually 1 – 2; values greater than 2 are rare (**Figure 11C**).





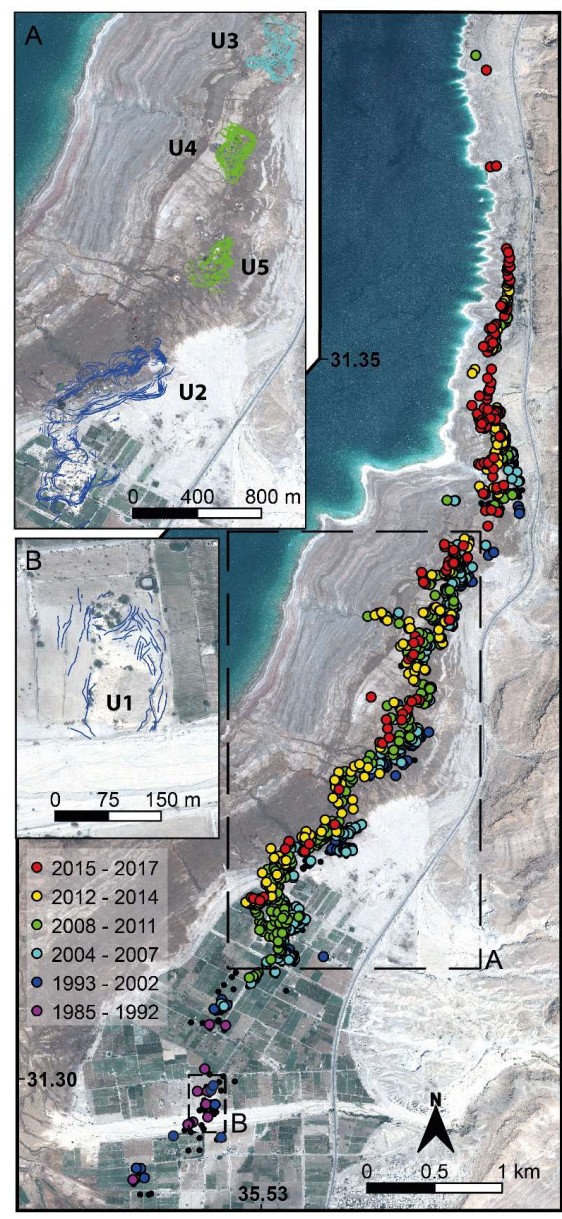




**Figure 9: Sinkholes mapped from satellite and aerial imagery, colour-coded by year of first sighting in time intervals as labelled. Base image is Pleiades 2017. Smaller black dots are sinkholes mapped prior to 2009 by other sources but not visible in our imagery. Insets A and B show the larger-scale depressions, as denoted by mapped ground cracks and fractures. For clarity, all fractures for each depression are colour-coded by year of first sighting of any fractures related to that depression. In detail, the fracture formation**
**ages in each large-scale depression span a greater range than shown here (see Figures 12 and 13).**

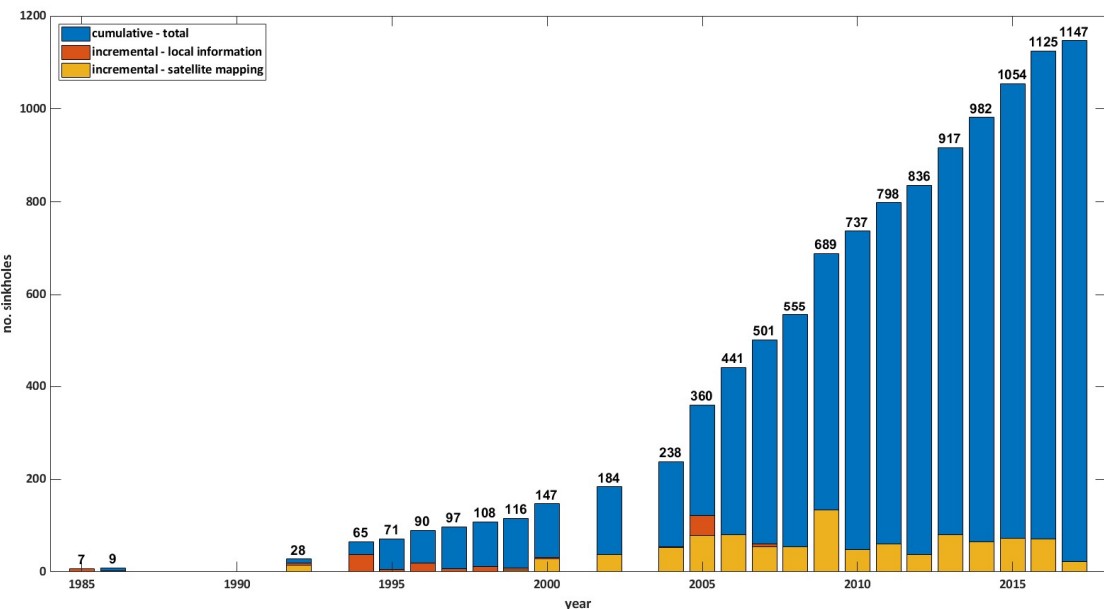

**Figure 10: sinkhole population growth with time in the study area. 'Local information' constitutes sinkholes noted in field surveys along with information provided by farmers on sinkholes that were filled in before they were mapped. The total number of sinkholes**
**mapped from satellite imagery alone is 996. The year 2017 appears to have a reduced number of new holes: this is a sampling artefact as there is only a four-month time interval between the 2016 orthophoto and the 2017 satellite image.**

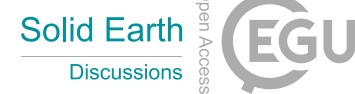

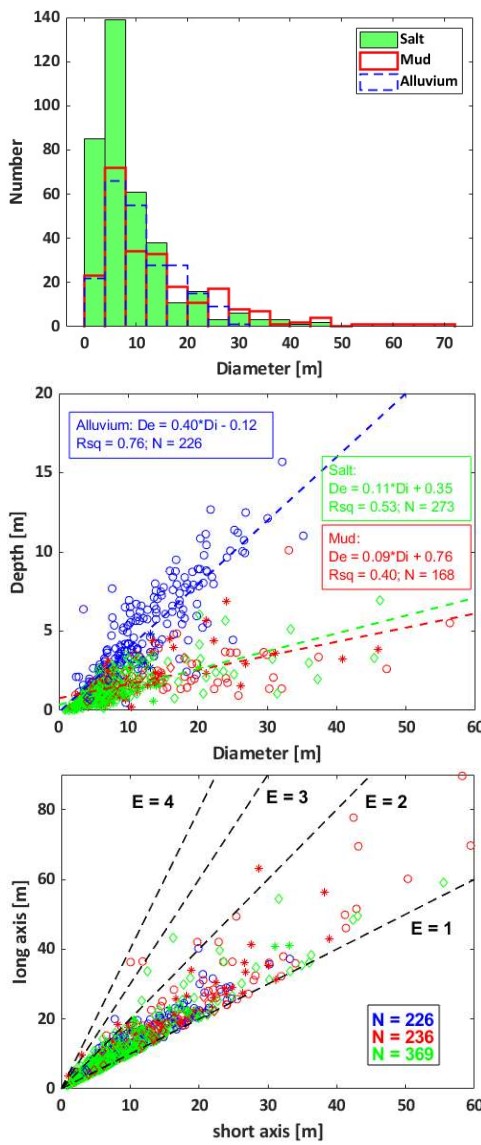


**Figure 11: Morphological characteristics of the sinkhole population developed at Ghor al-Haditha in various sedimentary materials. (A) Number of holes binned according to average diameter, (B) the relationship between depth and diameter, and (C) plan-view eccentricity (longest/shortest diameter). The total number of holes analysed is 226 in the alluvium, 236 in the mud-dominated**

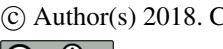



lacustrine deposits ('mud') and 369 in the salt-dominated deposits ('salt'). Water-filled holes were excluded from the depth/diameter
       analysis. The De/Di plot presents the maximum elevation difference between the rim and the bottom of each sinkhole.

### 4.3.2    Uvalas

Many sinkholes in the study area have developed within *uvalas* - gentle depressions of several hundred

metres in lateral extent (De/Di ratios = 0.015 – 0.033) (**Figures 9, 12 & 13**). The uvalas are bounded partly by systems of ground cracks and/or by faults with vertical displacements of up to 1.5 m. The expression of such fractures is material dependent. Subsidence-related displacements are accommodated on fewer but larger fractures in alluvium, whereas numerous but smaller fractures occur in mud-rich deposits. As shown below, these fractures are spatially and temporally associated with subsidence of each

uvala. They are not to be confused with regional tectonic structures.

Development of each uvala follows precursory sinkhole formation at that site. 2 – 8 years after the first sinkhole sighting (in which time many sinkholes have generally clustered about the initial hole), ground cracks develop that no longer trend concentrically to any single sinkhole, but instead delineate a wider

zone of subsidence that envelopes several sinkholes or even several clusters of sinkholes. The first uvala, U1, developed between 1992 and 1999 in the south of the area, near the Wadi Ibn Hamad (**Figure 9**). U2 and U3 initiated in 2002 and 2005-2006, respectively, to the north east of U1. Both U4 and U5 began forming around 2008, but lie between U2 and U3. Groundcrack patterns observed in the 2015 and 2016 orthophotos between Fans 1 and 2 suggest that a new uvala may develop further northeast of U3. In

general therefore, younger uvalas have formed to the northeast, as seen for the sinkholes, although not (yet) in as clear a sequence.

After initiation, uvala growth is closely linked with further sinkhole formation within it. For instance, groundcracks related to U2 initiated around two spatially-discrete sinkhole clusters; these fractures sets

propagated and joined as sinkhole development migrated (**Figure 12**). For both U3 and U4, two 'prongs' of coeval crack and sinkhole development are visible (**Figures 12 and 13**). Conversely, U1 ceased development by 2006, in tandem with diminshed sinkhole activity nearby. Similar to the sinkhole





population growth trends, the expansions of the uvalas U2, U3 and U4 has been mostly seaward and accelerated markedly in 2008-2009 (**Figure 14**).


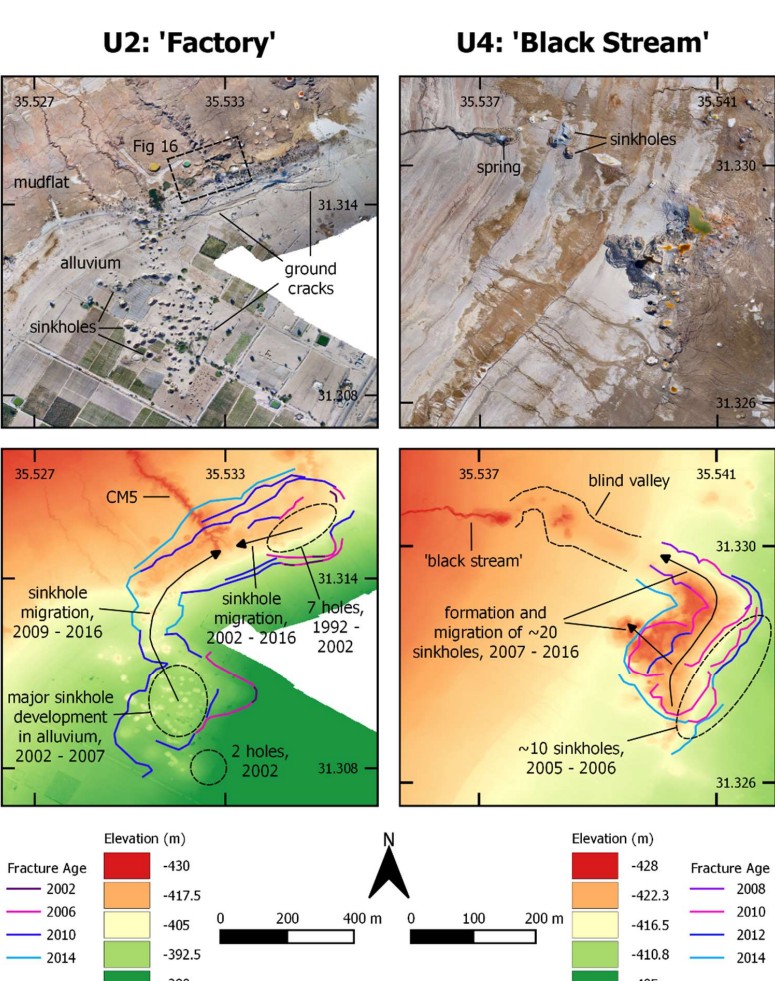

**Figure 12: Structural development of uvalas with demonstrable connection to channelized subsurface water flow,. See figure 9 for locations. The years of formation of the main depression-bounding fractures (i.e. when first visible in imagery) are grouped and coloured in four-year intervals for U2 and in two-year intervals for U4. Each uvala is linked morphologically to a highly active stream that emerges on the seaward side at several meters below the surrounding ground surface.**




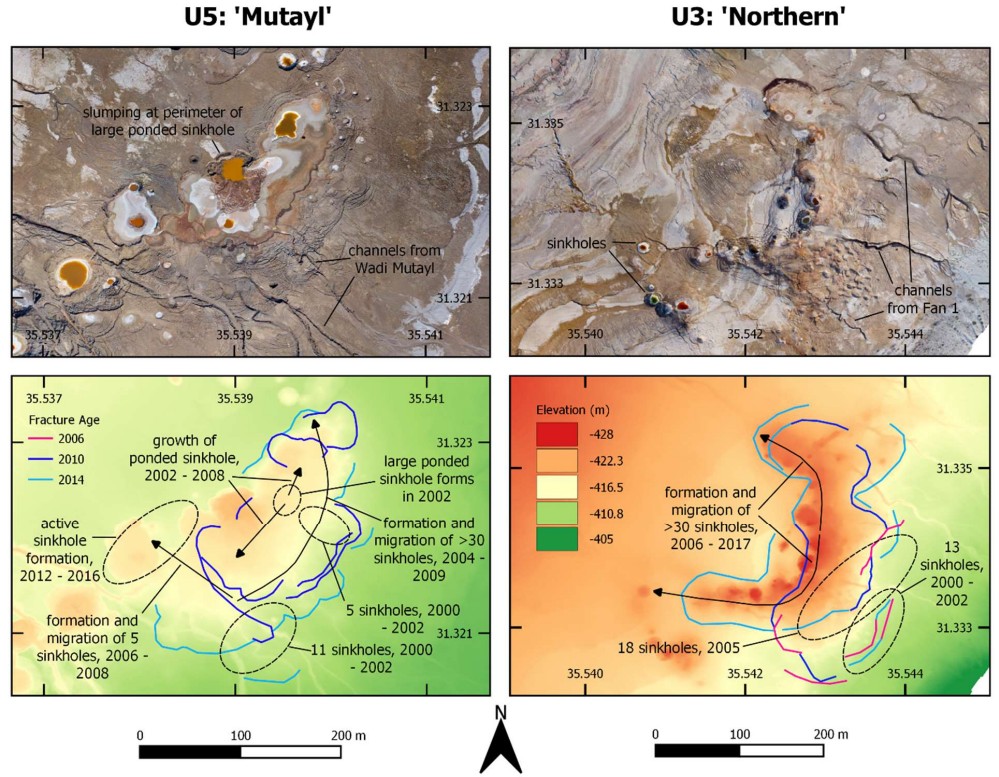

**Figure 13: Structural development of uvalas U5 and U3 with unclear connection to subsurface water flow. The years of formation of the main depression-bounding fractures (i.e. when first visible in imagery) are coloured in four-year intervals.**


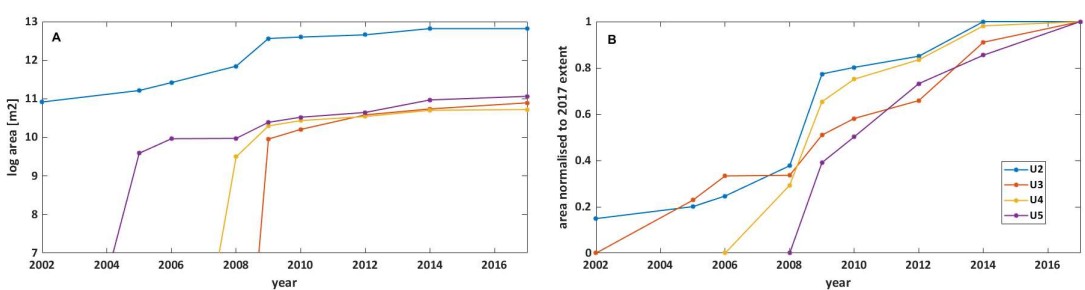

**Figure 14: Growth of uvalas in the study area, shown as (A) absolute areas and their (B) area normalised to that in 2017.**





### 4.4 Notes on properties of surface water and groundwater

#### 4.4.1 Springs and streams

Springs within the Ghor al-Haditha sinkhole area can be categorised as sulphurous or non-sulphurous (**Figure 2A**). The water temperatures of both spring types were between 27.7-30.6 °C. Non-sulphurous springs emerge from the Ajlun Group aquifer (**Figure 15A**) and from near the alluvial/lacustrine sediment boundary in the south of the area (**Figure 15B**). Sulphurous springs are most common in the north of the area, where they emerge from the salt-rich deposits, and are characterised by dark, turbid water, with gas

bubbles and a strong smell of $H_2S$ (**Figure 15B**). Sediments surrounding such springs commonly show a black, green or reddish staining. Sulphurous springs have also been recorded historically in the adjacent outcrops of the Ram-Kurnub aquifer (**Figure 2A**) (Khalil, 1992).

Electrical conductivity values of the spring waters show an extremely wide range (1-217 mS), but a well-

defined spatial distribution (**Figure 15D**). Conductivity values of springs emerging from Ajlun Group bedrock are 1-14 mS, while values of 13-78 mS characterise streams emerging in the mud-rich sediments on the seaward side of the sinkhole-affected area (**Figure 2A**). The highest conductivity values – some higher than the value of the Dead Sea itself (180 mS) - were measured for sulphurous springs emerging in the salt-rich sediments in the north of the area (**Figure 2A**). The $\delta^2H$ and $\delta^{18}O$ composition of all

spring waters plot below the local recharge values, which are marked by the Eastern Meditarranean Meteoric Water Line (EMMWL) (**Figure 15E**). Fresh to brackish groundwaters from the Ajlun Group and the superficial alluvium aquifer are isotopically lightest, referring to regional recharge that experienced slight evaporation before infiltration. Sulphurous springs are isotopically heavier, possibly due to admixture of highly enriched interstitial brines hosted within the salt-flat sediments (cf. Siebert et

al., 2014).

Discharge rates of the largest active streams were estimated near their source springs during the 2015 field campaign. These included the main stream in CM5 (discharge = 0.20 m³/s) near the destroyed factory site, the 'black stream' in CM7 (discharge = 0.07 m³/s) emerging in the centre of the area and the stream





in CM8 (discharge = 0.04 m$^3$/s) in the north of the study area. Most of the other channels in the area were of much lower discharge or had no discharge during field campaigns in 2014-2016.

### 4.4.2 Ponds

Ponds within sinkholes range markedly in water colour, organic content (algal/bacterical scum) and salt rim development. The electrical conductivity of ponds (47-218 mS) is generally higher than the springs
(**Figure 15D**). Highest values were recorded for ponds in salt-rich sediments, but values varied greatly even between adjacent ponds of similar appearance (**Figure 2A**). Pond water temperatures ranged from 26.0-32.4 °C. $\delta^2$H and $\delta^{18}$O signatures in pond brines are heaviest among all observed fluids, mainly due to strong admixture of interstitial brines and due to high evaporation within ponds. Brines in mud-edged ponds represent the isotopic heaviest fluids in a mixing line (ML in **Figure 15E**) between fresh/brackish
groundwaters (green in **Figure 15E**) and interstitial brines. However, isotopic signatures in brines stored in salt-edged ponds deviate from that mixing line due to much higher evaporation, leading to higher salinity and subsequent precipitation of the observable salts. The overall trend is consistent with effect of evaporative fractionation, and the line (ET in **Figure 15E**) that best fits the data ($R^2 = 0.9$) has a slope of 3.9, which is consistent with such fractionation under arid conditions (relative humidity < 25%).




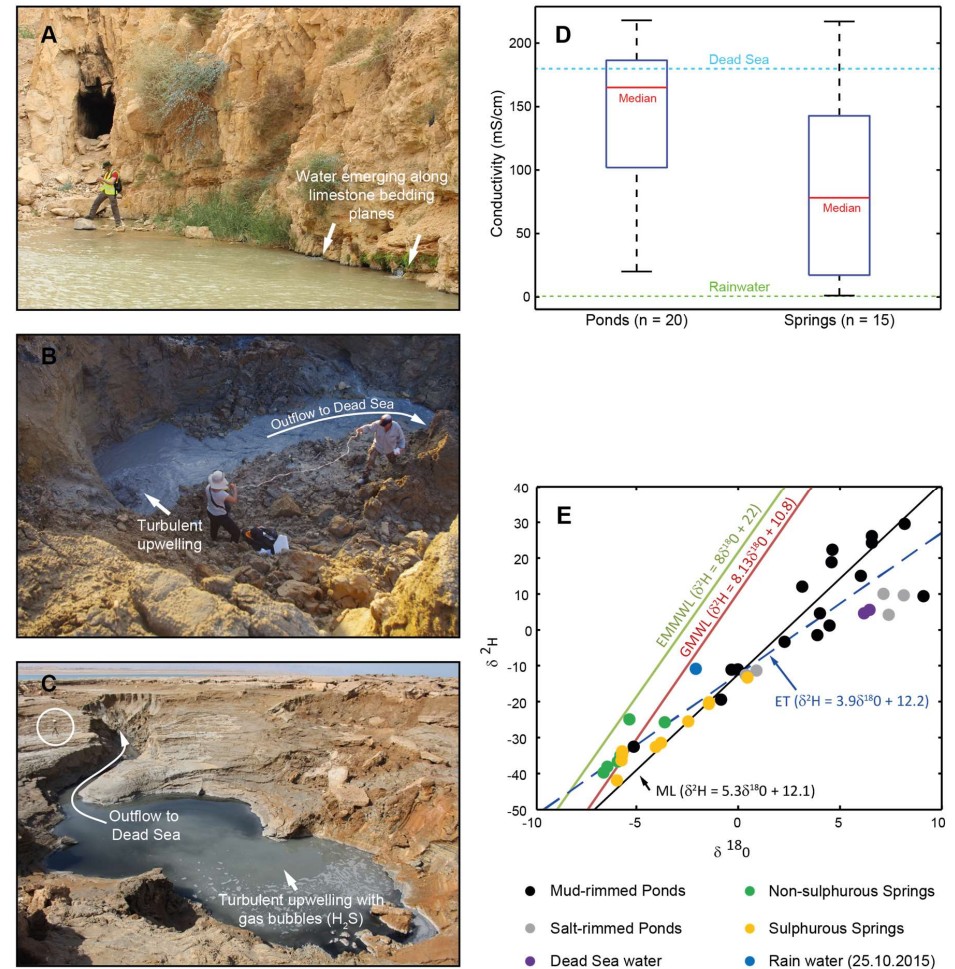

**Figure 15: Springs types and electrical conductivity measurements in the Ghor al-Haditha area. (A) Non-sulphurous Al-Maghara ('the cave') spring in the Wadi Ibn Hamad. White arrows point to water flowing from bedding planes in the limestone of the Aijun bedrock aquifer. Person for scale. View is to the east. (B) Non-sulphurous spring formed between Oct. 2014 and Oct. 2015 at the head of channel CM5 (see also Figures 1 and 4) in the mud-rich lacustrine deposits. Grey-lilac colour is from suspended mud load. The spring lies about 5 m below the top of the lacustrine deposits, from where the image was taken. (C) Sulphurous spring at the head of Channel CM6 ('Black stream') in salt-rich lacustrine deposits. A 35 cm high GPS tripod is circled for scale. The spring lies about 2-3 m below the former lake bed. The water colour is cloudy, dark-grey or blackish. The froth at the surface is from gas ($H_2S$) bubbles. (D) Electrical conductivity of ponds (water-containing sinkholes) and springs in the Ghor al-Haditha area. Also shown for reference are the conductivities of local rainwater and the Dead Sea water. (E) $\delta^2H$ and $\delta^{18}O$ data for springs and water-containing sinkholes in the Ghor al-Haditha area. Eastern Mediterranean Meteoric Water Line (EMMWL) from Gat and Dansgaard (1972) and Global Meteoric Water Line (GMWL) taken from Dansgaard (1964). Mixing Line (ML) and evaporation trend (ET) line.**



## 5    Discussion

### 5.1    Base-level fall and shoreline retreat

Our results as presented above clearly demonstrate that the evolution of various geomorphological phenomena in the Ghor al-Haditha area, which include new fluvial channels and alluvial fans, as well as new sinkholes and uvalas, are intrinsically linked to a common driver: the fall in regional hydrological base level resulting from the drying of the Dead Sea. In particular, the base level fall has exposed a large area of the former Dead Sea bed to fluvial erosion and deposition as a consequence of shoreline retreat.

The spatiotemporal variation in the rates of retreat result from the temporal variation in the rate of base-level fall and, more importantly, the spatial variation in the local bathymetry (**Figure 3**). The northern part of the study area had a steeper bathymetry, reflecting its proximity to the bounding fault scarp of the Dead Sea basin (**Figure 2A**). The southern part of the area had a gentler bathymetry, possibly reflecting the extensive Plio-Pleistocene and Holocene fan deposition at the terminations of several major wadis

(**Figure 1C**). This variation in bathymetry and rate of shoreline retreat represents the framework for the past and the future geomorphological changes.

### 5.2    Geomorphic characteristics of new stream channels in cohesive lacustrine materials

At Ghor al-Haditha, channels that formed display distinct map-view morphologies: meandering and straight/braided. Such differences in morphology result from a complexly interacting set of factors,

including (i) the flow discharge, (ii) the slope prior to channel formation (iii) the sediment load characteristic and (iv) the properties of the material into which the channel is cut (Buffington and Montgomery, 2013; Leopold and Wolman, 1957; Schumm and Khan, 1972). Since the Ghor al-Haditha channels have formed in the same cohesive lacustrine sediments and along similar bathymetric profiles, we deduce that the observed differences in map-view morphology arise from differences in discharge rate

and sediment load. Specifically, we infer that the meandering channel geometries stem from steady, low discharge of groundwater from springs and a silty clay sediment load. In contrast, we infer that the straight channels are fed by episodic higher-discharge events (flash floods) from nearby wadis bearing higher volume and coarser sediment loads. Such inferences are supported by our field observations of sediment deposits within each channel type.





In cross-section, the two channel types show v-shaped profiles with average W/D ratios of 5 - 15 (**Figures 5 & 6**). Such W/D ratios are thought to be typical of single thread or anastomosed channels in cohesive materials (Church, 2006; Simon and Darby, 1997), although field studies of channels in such materials are rare (Vachtman and Laronne, 2013). Similar but more tightly constrained W/D ratios of 12-15 are reported for rivers with clay-dominant substrates by Ebisa Fola and Rennie (2010). The straight channels

(CS1-6) show some higher W/D ratios (20 – 40) in their downstream sections close to the shore, however. These channel sections correspond to areas of lowest slope (**Figure 6B**), and so could be influenced by that factor. Another factor, possibly complementary, is that landward growth toward the alluvial fan (**Figure 4**) had facilitated greater discharges and sediment loads by the time these lower sections formed.

### 5.3    Geomorphic characteristics of new stream channels in non-cohesive alluvium

In contrast to the v-shaped channel cross-sections in the mudflats, trapezoidal cross sections observed in the marginal alluvium (**Figure 7**), which match well with observations on the western shore of the Dead Sea (Bowman et al., 2010). The cross-section profiles and the straight/braided map-view geometry of these channels are consistent with controls of relatively high discharge vs slope (Leopold and Wolman, 1957), and of low substrate cohesion (Schumm and Khan, 1972; Peakall et al., 2007).


Channel incision into the old marginal alluvium represents a hazard to infrastructure - specifically the adjacent north-south highway along the Dead Sea's eastern shore (**Figure 7**). Constructed in the early 1990's, the road section lies close to, and in places seaward of, the 1967 shoreline. Reinforced concrete bridges and drains were emplaced at intersections with wadis to guide stream water under the road. The

foundations of some of these bridges now lie at knickpoints in the stream profiles, with enhanced erosion at their downstream sides. The future development of such erosion and bridge integrity should be monitored closely.





### 5.4 Effect of base level fall on channel morphology in space and time

The most obvious effect of base level fall on the channel development at Ghor al-Haditha is the progressive seaward incision of newer channel segments into the lacustrine deposits as the shoreline retreats over time (**Figure 4**). Coeval with incision of new channel sections, we also observe progressive widening of existing upstream sections. For the meandering channels, the relatively constant W/D ratios along the present channel profiles (**Figure 6A**) show that these older channel sections must have also

deepened with time. Finally, the gradual decrease in sinuosity seen for older channel sections (**Figure 6C**), which is independent of slope (**Figure 6D**), suggests that sinuosity has also progressively decreased with time. The persistence of such changes in time and space in such small scale channels suggests that the continuous base level fall has inhibited the development of equilibrium channel geometries (cf. Simon and Darby, 1997).


The pattern of sinuosity variation along the channel length (**Figure 6C**) is compatible to some degree with conceptual model predictions for effects of incremental base level fall. Schumm (1993) proposed an increase in sinuosity only in the lowermost channel sections for an increment of base level fall. This concept is agreement with the observed sharp increase in sinuosity immediately up stream of the channel

mouths (**Figure 6C**). Subsequent continuous base level fall may account for the observed gradual decrease in sinuosity further upstream, as (1) the lowermost channel section progressively migrates seaward and (2) overall channel slope increases (Yoxall, 1969).

Head-ward adjustment differs for each channel type, however. The heads of straight channels migrate

landward (upslope); the heads of meandering channels deepen rather than migrate. The upslope incision and migration of the channel head in the straight channels is consistent with experiment results from Koss et al. (1994), in which base level fall occurred across a substrate with a marked change of gradient. As base level in the experiment fell, the channels developed at a transition from gentle to steeper slope and then progressively deepened and eroded headward. A similar variation in slope geometry is present

between the Wadi Mutayl fan and the area where heads of straight channel CS1-3 initially formed (compare **Figures 3 and 4**). In contrast, the heads of most meandering channels are fixed in space and



time (**Figure 4**), simply because the power of subsurface flow (a function of slope and discharge) inland of the spring points is insufficient to destabilise the overburden there. An exception to this behaviour is seen in channel CM5 (**Figures 4 and 16**), which has the highest measured discharge of all the meandering
channels.

### 5.5    Sinkhole morphology and spatial distribution

The contrast in morphology of sinkholes formed in alluvium or mud-rich lacustrine materials at Ghor al-Haditha (**Figure 11A, B**) is also observed elsewhere around the Dead Sea (Filin et al., 2011; Al-Halbouni et al., 2017). The lower De/Di ratios of the mud-hosted sinkholes and their longer tail in size distribution
toward diameters greater than 70 m has been attributed to contrast in strength (Al-Halbouni et al., 2018) and/or rheology (Shalev and Lyakhovsky, 2012) of these materials. The high strength and/or frictional rheology of the alluvium inhibits lateral expansion of the sinkhole as it deepens, whereas the low strength and/or viscoelastic rheology of the mud-rich sediments enables lateral expansion while reducing the depth.


Expanding upon data presented by previous authors (cf. Filin et al., 2011; Al-Halbouni et al., 2017), we show that De/Di ratios of sinkholes formed in salt-rich lacustrine sediments generally fall between those of sinkholes formed in alluvium or mud-rich lacustrine sediments (**Figure 11B**). Compared to those in mud-rich sediments or alluvium, sinkholes formed in the salt-rich sediments also have generally smaller
diameters (**Figure 11A**). This could reflect a scaling limit imposed by the level of karstification, which in the salt-rich material in the northern part of the area is at, or within a few metres of, the surface.

At the kilometre scale, the spatial distribution of sinkholes at Ghor al-Haditha follows two linear trends: a N24° trend in the south and a N10° in the north (**Figure 9**). These trends match those of main regional
faults in the Dead Sea transform (**Figure 1A**) and so indicate a tectonic control on overall sinkhole distribution (cf. Closson and Abou Karaki, 2009). Similar tectonic controls have been suggested on the spatial distribution of sinkholes on the Dead Sea's western shore (Abelson et al., 2003; Yechieli et al., 2016). In closer detail (hundred-metre scale), the sinkhole distribution is non-linear or sinuous (**Figure**





**9**). This non-linearity may reflect control from the distribution of salt-rich deposits at depth (Ezersky et

al., 2013), and thus reflect the palaeo-shoreline, as determined by the regional fault systems.

### 5.6    Uvalas

The uvalas at Ghor al-Haditha (**Figures 9, 12 and 13**) are distinct from the sinkholes in terms of their scale and morphology. The uvalas are much more irregular in plan-view and have De/Di ratios an order of magnitude lower. Their irregular shape indicates material removal in a 'linear' or 'areal' sense as

opposed to shape being governed by material removal at a point, as is the case with sinkholes.

The exact definition of uvalas and the processes contributing to their formation are still debated in karst geomorphology (Ćalić, 2011; Kranjc, 2013; Lowe and Waltham, 1995). The shallow but laterally extensive morphologies of the salt-karst uvalas at Ghor al-Haditha agree well with Ćalić (2011)'s

observations of uvalas in shallow limestone karst where the water table is close to the base of the depression, which causes the characteristic 'widening without deepening' evolutionary pattern. Uvalas are distinct from *poljes* (the second most diagnostic karstic depression after *dolines*, sometimes considered to be the karstic equivalent to a fluvial valley) in that that their bottom is always situated above the karst water table and is generally more undulating and often pitted with sinkholes (Ćalić, 2011).


Our observations at Ghor al-Haditha provide new insight into the development of salt-karst uvalas. There is a clear spatiotemporal link between initial sinkhole clusters and the uvala formation (**Figures 9, 12 and 13**). In all cases, some precursory sinkhole development occurred several years before. The development of the uvalas after this precursory sinkhole formation is linked with further sinkhole formation. Uvala

formation and sinkhole developemnt initiate develop and cease in tandem, thus indicating the same overall formation process.

### 5.7    Effects of base-level fall on sinkhole development

A striking observation from our study is that, similar to the fluvial channels, sinkhole clusters consistently show a seaward growth after they have been established (**Figure 9**). Several past studies on the Dead Sea



sinkhole problem have postulated that the fall of base level should affect the interface developed between the hypersaline Dead Sea brine and less saline, brackish (i.e. relatively 'fresh') groundwater (Salameh and El-Naser, 2000; Yechieli, 2000; Yechieli et al., 2009) (**Figure 2B**). In theory, this 'fresh-saline interface' should shift seaward in tandem with the retreating shoreline, enabling groundwater undersaturated with respect to halite to infiltrate the salt-rich deposits in the subsurface, thus triggering

karstification and surface subsidence. A prediction of this theory is that karstification and new sinkhole development should shift seaward also, although evidence on the western shore for such shift is weak (Abelson et al., 2017; Charrach, 2018). Although we lack constraints on the fresh-saline interface from sources boreholes or geophysical techniques in the Ghor al-Haditha study area, our observations of systematic shoreward sinkhole migration provide the strongest evidence yet that seaward shift of the

fresh-saline interface induced by base-level fall can be a key control on sinkhole development.

In detail, the rates of shoreward migration of sinkhole development have been variable in space and time, and this may relate to other local or transient controls. The salt-rich evaporite materials may be anisotropically distributed through the subsurface in the study area (Polom et al., 2018). Focussing of

groundwater into salt-rich 'lenses' in given years could explain the rapid development of new conduits at those times, hence leading to locally accelerated migration (**Figure 9**) and to jumps in the growth of the sinkhole population and of the uvalas, such as occurred in 2005 – 2006 and 2009 (**Figures 10 and 14**). Alternatively, or perhaps complementarily, such small-scale and short-term fluctuation in the spatiotemporal development of the sinkholes at Ghor al-Haditha may relate to variations in meteoric

and/or groundwater inflow as inferred for several sinkhole sites on the western shore (Abelson et al., 2017).

### 5.8    Links between surface stream flow, subsurface stream flow and sinkhole formation

Several features of the uvalas and the formation of sinkholes within them strongly suggest a link between their development and the channelized flow of groundwater. The best example of such links is seen at

uvala U2 in association with the development of stream channel CM5 (**Figures 4, 12 and 16**). Firstly, CM5 developed from a spring, which emerged in the middle of the mudflat and in association with



drainage of a lake hosted in U2 (**Figure 4**). Secondly, the migration patterns of sinkholes within U2 converge at the spring location (**Figure 12**). Thirdly, upstream incision at the head of CM5 is spatially and temporally linked with sinkhole collapses (**Figure 16**), which occurred on a time-scale of a few days.

These collapses show that the water reaches the head of CM5 via subsurface conduits. Fourthly, U2 also hosts a vegetated pond (labelled $s_1$ in **Figure 16**), which is fed by fresh non-sulphurous groundwater. This water therefore passes through the alluvial sediments without being significantly salinized by interstitial brine, indicated by low salinity (EC=20 mS/cm) and light $\delta^2$H and $\delta^{18}$O signatures. These observations indicate the presence of subterranean channel flow, followed by surface flow and channel incision.

Therefore, in compliment to the arguments presented by Avni et al. (2016), the rapid development of sinkholes may not only occur due to 'flash-flood' style input, but also from steadier groundwater input.

Further evidence of such links is seen at Uvala U4, which is linked spatially and temporally with the spring feeding channel CM7 (the 'black stream') (**Figure 12**). Initial ground cracking at U4 occurred

proximal to a subtle linear depression (or 'blind valley') between it and the spring feeding the 'black stream', which is first observed in 2009. The migration of sinkholes within U4 seems to follow a flow path from the intial pre-uvala sinkhole cluster to the 'black stream' head, suggesting the presence of a flow conduit beneath the depression. Additional 'prongs' of sinkhole migration and groudcracking at uvalas U4 and U3 (**Figures 12 and 13**) may also represent a suface expression of subsurface conduit

development, although no associated springs have yet been observed.

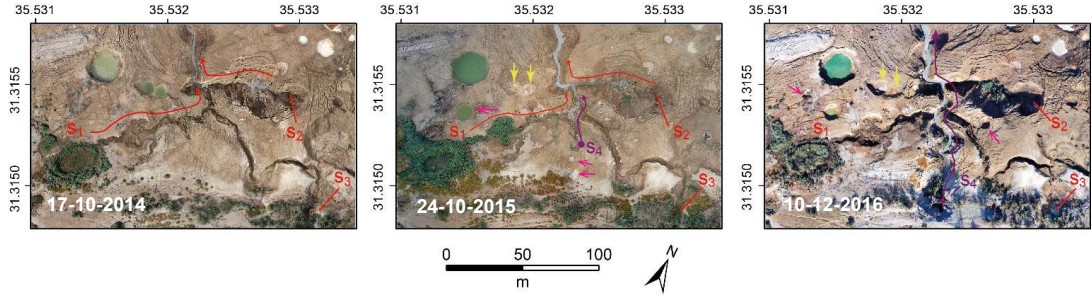

**Figure 16: Orthophotos of the canyon-spring-sinkhole system close to the former mud-factory at the head of CM5, displaying the evolution of surface water flow and sinkhole collapse and their links to subsurface flow. In 2014, $s_1$ and $s_2$ both feed downstream CM5. By 2015, a new spring, $s_4$ has formed and cut back to the southeast, and carries more water than $s_1$ and $s_2$. New sinkholes have**

**formed nearby (green arrows). The formerly ponded sinkholes (labelled with yellow arrows) have dried considerably. By 2016, $s_4$**

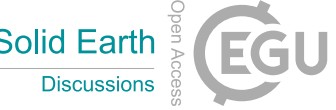

carries all of the flow to CM5, and has back-eroded further. More new sinkholes have formed, and s$_1$ and s$_2$ are now totally dry. The spring that originally fed CM5, labelled s$_3$, has been dry since 2013.

These links between surface water, groundwater and sinkhole development show that the upper limit of
subsurface karstification is variable across the study area but overall very shallow. Karstic development within the salt-rich deposits proximal to CM8 (**Figure 1**) was observed at less than a metre beneath the surface in the field (Al-Halbouni et al., 2017). Close to CM7 (the 'black stream'), subsurface channelization must occur at a minimum depth of around -428 m (**Figure 12**) (the base of the adjacent uvala U4 is close to this elevation). Further south, evolution of CM5 and related sinkhole collapses
shows that subsurface channelization occurred at less than 10m below the surface. The lower elevation limit of karstifcation is unknown but seismic and borehole data indicate that it locally must be at depths greater than 45 – 100 m (Polom et al., 2018).

Finally, the origins of the groundwaters driving the subsurface karstifcation and much of the surface
channel development appear from our preliminary surveys to be sourced from all three of the adjacent aquifers (**Figures 2A and 15**). Sulphurous springs linked to sinkhole and uvala formation in the north of the area lie downslope of the Ram-Kurnub aquifer which is know to emit reducing sulphurous water (Charrach, 2018). In contrast, non-sulphurous springs are linked with sinkhole and uvala develoment in the south of the area, and are likely sourced from the adjacent Aljun and superficial gravel aquifers.

**5.9    Limitations and future work**

Much of the available data on the study area's 50-year evolution is necessarily 2D in nature, as our 2014-2016 photogrammetric surveys were the first to yield 3D data at sufficient resolution. Additionally some of our analysis is necessarily qualitative in the absence of quantitative constraints on past stream discharge rates and groundwater levels. Future appraisal of the geomorphological evolution of this area and the
related hazards ideally requires the following: (1) additional high-resolution and high-precision 3D topographical surveys, ideally at annual frequency or better; (2) monitoring of stream discharge and sediment load at high temporal resolution; (3) systematic drilling of boreholes to constrain subsurface lithologies, groundwater properties and groundwater levels.



## 6    Summary & Conclusions

Our results provide, for the first time in the study area, a detailed picture of the interlinked fluvial and karstic geomorphological responses of surface and subsurface hydrological systems to base level fall at the Dead Sea. Our main findings are as follows:

(1)    The continuous and rapid fall in base level of the Dead Sea over the last 50 years has resulted in a

635        number of geomorphological changes on the eastern shore, including: (i) incision of fluvial channels of atypical morphology into the former sea bed and into the marginal alluvial fan deposits; (ii) growth and progradation of new alluvial fans, (iii) formation of many sinkholes and several salt-karst uvalas.

(2)    Channel morphologies in the former lakebed are V-shaped in cross-sectional profile and in plan-view can be divided in to meandering and straight types. Meandering channels show low W/D ratios (5-

640        15) that are consistent along the longitudinal profile of the channel. These channels also show relatively low sinuosity (1.1 – 1.7). Straight channels show similar W/D ratios in their upper reaches but higher ratios (15-40) in their lower reaches, which are commonly braided. Channel morphology in the marginal alluvium is U-shaped (trapezoidal) in cross-section profile and straight in plan-view.

(3)    The differences in channel morphology in the former lakebed are primarily related to differences in

645        discharge and sediment load. Water in meandering channels is sourced from low-discharge groundwater springs and carries low-volume clay/silt dominated loads. Water in straight channels is probably sourced from flash-flood events traversing adjacent alluvial fans and carries higher volume loads of sand to cobble grade.

(4)    Consistent with experimental studies, the relatively low sinuosity and low W/D of the channels in the

650        former lakebed are compatible with a combination of the cohesive nature of the channel substrate and with a forcing by the continuous rapid base level fall.  The factors inhibit lateral erosion and channel migration, and so vertical incision is the dominant response to base level fall.

(5)    Over 1100 sinkholes have developed at Ghor al-Haditha since the mid-1980s. Rate of formation of sinkholes accelerated from the 1980s until 2009; since then a lower but steadier rate of formation

655        (~60 holes per year). Sinkholes at Ghor al-Haditha form and grow in clusters. New clusters have initiated from SSW – NNE, roughly parallel to the modern shoreline.





(6) Sinkhole morphology is variable depending upon the material properties of the lithology in which they formed. Sinkholes formed in alluvium tend to have the highest De/Di ratios (average 0.40), while those formed in mud-rich lacustrine deposits tend to have the lowest De/Di ratios (average 0.09). Sinkholes formed in mud-rich lacustrine deposits can achieve much larger diameters than those in alluvium or salt-rich deposits.

(7) Several salt-karst uvalas have developed around and in tandem with clusters of sinkholes. These uvalas are areas of subsidence several hundreds of metres in scale and in part bound systems of ground cracks and faults.

(8) The main geohazards arising from base level fall are fluvial erosion and sinkhole formation. Although there is still some risk to infrastructure in the more densely populated areas to the south, these hazards pose the greatest future risk to the infrastructure (Dead Sea highway) in the north of the study area.

(9) A progressive seaward migration in the development of the stream channels, alluvial fans, sinkholes and uvalas at Ghor al-Haditha ultimately reflect the underlying effect of base level fall on fluvial and karst systems. Seaward growth of channel and alluvial fans reflects incision into and deposition onto the progressively uncovered lakebed as the shore line retreats. Similarly, consistent seaward migration of sinkhole and uvala development may be linked to seaward propagation of a dissolution front, facilitated by lateral retreat of the fresh-saline interface at depth as the base level falls.

## 7    Data Availability

A full set of metadata is available upon request. Satellite images: some open access (Corona), but mostly commercial. Aerial images: available at discretion of RJGC. Photogrammetric surveys: raw images, DSMs and orthophotos available upon consultation with the authors. Geological Map 1:50,000 Ar Rabba: available at discretion of MEMR.

## 8    Author Contribution

RAW and EPH led the production of figures and writing of the manuscript. RAW undertook the majority of the data analysis associated with the satellite imagery time series and the 2015 and 2016 DSMs.





Additional satellite imagery processing and data analysis was performed by LS, DAH and EPH. DAH and LS generated the orthophotos and DSMs of the study area using SfM photogrammetry. EPH, DAH, LS, HAR, and AS undertook the field studies and close-range photogrammetric surveys in 2014 – 2016.

Water sampling was performed by EPH and LS in 2015, and isotopic analysis was overseen by CS. All authors reviewed and commented on the manuscript, and they contributed to discussions of the data.

## 9    Competing interests

The authors declare that they have no conflict of interest.

## 10    Special issue statement (will be included by Copernicus)


## 11    Acknowledgements

We acknowledge MEMR colleagues for support in fieldwork and other logistical support. The authors acknowledge financial support from GFZ and the Helmholtz Association's recent Dead Sea Research Venue (DESERVE) initiative (Kottmeier et al., 2016), especially for the associated data and fieldwork

costs. Part of the work of NAK was done during a sabbatical year supported by the Deanship of scientific research – The University of Jordan. RAW and EPH are also grateful to the Geological Survey of Ireland for providing funding for RAW's masters research project, of which this manuscript forms a central part.

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
