# Peer review of "Sinkholes and uvalas in evaporite karst: spatio-temporal development with links to base-level fall on the eastern shore of the Dead Sea"

_Solid Earth, 2018_

## Referee Comment (RC1) · Anonymous Referee #1 · 19 Nov 2018

General: The manuscript shows a detailed work on geomorphological processes related to the Dead Sea base-drop for the last 50 years as they express in alluvial incision and sinkholes formation at Ghor al-Haditha, Jordan. The methods include topographic analysis, orthophoto analysis, hydrological isotopes analysis and field observations. The research shows some findings of channel morphology as expected from the local slope, strata and inflow and sinkhole formation in accordance with previous findings along the western Dead Sea shores. A major missing component (as admitted by the authors) is an analysis of the hydrological boundary of the fresh-saline water in particular and the underground water levels and composition in general. In addition, the authors fail to properly contextualize the results with previous findings along the Dead

[Figure]

Sea area. The manuscript and the readers will benefit from detailed comparison with similar results described in the papers already cited in the current manuscript. Furthermore, the isotopic analysis is incomplete, its results are faintly included in the discussion and cannot support any conclusion regarding salinity. Na/Cl content for example would provide more conclusive information regarding dissolution processes. Overall, I find this paper findings to be of much interest showing the Dead Sea base-drop effects are similar on similar environments on either side of the Dead Sea. However, in its current form, I find it is more of a summary of observations and has limited scientific value. I would suggest a major revision and addressing the points below:

specific comments: Line 31: The response of the surface and subsurface hydrological systems to the base-level drop have been presented previously by e.g. Arakin et al., 2000 env. geol.; Bowman et al., 2007, Geomorphology; Avni et al., 2015, JGR; Shviro et. al., 2017, Geomorphology; I would suggest avoiding using the term "first" here, or explain in detail this research novelty in this context. Line 142: Some error estimations should be provided for the co-registration as done for the DSMs. I'm concerned 9 GCPs are not enough for proper geocoding. Line 189: Please add a theoretical line, based on water level drop and slope. I suspect the non-linearity origin is from the non-linearity of the water-level drop rates. As it is described now, one might think it is an abnormal observation, while it might be an expected one. If it does not in agreement with the expected line, a more detail discussion should be added. Lines 291-299: I would suggest putting the sinkhole morphology in context with previous (similar) findings from the western Dead Sea shore. This will strengthen the globality of the findings and put them in proper context rather than highlighting a very local phenomena. Line 401: It is not clear why there should be higher evaporation in the salt-edge ponds with respect to mud-edge ponds? They are situated in very close proximity and same environmental conditions. Further water composition analysis would be useful for determine if water samples are of evaporative fractionation or mixing of different compounds. I suspect the difference between the two pond types is mainly due to salt dissolution. In addition, the isotopic result is not included in the

discussion and have little to no support to the conclusions. I would suggest expanding the isotopic and hydrochemistry analysis and to include it in the interpretation. An example of such analysis could be found in e.g. Avni et al., 2016. Line 438: In line 426 it is stated the northern part has steeper bathymetry and here that they are similar. Line 440: Discharge rates are only quoted for the meandering channels and no information is provided for flash floods. I fail to understand how sediment load is related to the morphology. Here you refer the sediment deposits only to support the assumption of the discharge rates. I would suggest obtaining estimations of flash floods discharges to support this assumption. Could the coarser sediments might be originally forming the channel beds and not transported by flesh-floods? Lines 512-513: These findings should also be discussed in context of Baer et al., 2018 (doi: 10.1002/2017JF004594) findings. Lines 532-536: The depth of the water table in the area and that of the Halit deposits (if present) are required to make this comparison between shallow limestone karst and the Dead Sea Uvalas. Without additional data, the depressions are "widening without deepening" where the base-level fall can be as easily explained by the fact that the karstic layer (Halit) is limited in its thickness as observed on the western side of the Dead Sea (e.g. Ychieli at al., 2016). Line 541: I fail to see the new insights here. The link between the Uvales formation and sinkhole process is documented in several pervious papers sited in the manuscript. Line 556: The statement "Evidence . . . is weak" is simply wrong. See for example Avni et al., 2016, figure 6. The seaward shift with time is much more pronounced that in the current paper. Line 559: I cannot see why this is a stronger evidence than that of e.g. Abelson et al., 2017. Without any information on the fresh-saline interface, it cannot support this theory. Channeling may explain the observations much as well (Arakin et al., 2000) without any evidence of a salt layer and dissolution processes. Line 564: The findings of Polom et al., 2018 of missing slat layer in the fan area may indicate a local area on increased fresh water streaming and accelerated dissolution that removed the salt layer in that area by the time of survey. These results, should be considered with much care for inferring general process related conclusions. The conductivity and mineral contents of the water

samples may indicate dissolution processes which is in contrast with Polom et al., findings. A more detailed hydrological analysis may better resolve this issue. The fact that with time, sinkhole distribution is along the whole area, (almost) without gaps, along a very distinct sub parallel line to the shore indicates the possible presence of an underlying salt layer undergoing dissolution processes. Line 583: technical corrections: Line 97: "there three" should be "there are three". In general to all figures with topographic data: I would suggest overlaying the color coded elevation over a hillshaded elevation to better express fine detailes. Line 144: Please add a proper citation to the GDAL library (see: https://github.com/OSGeo/gdal/blob/master/CITATION). Line 484: "is agreement" shold be changed to "is in agreement" Line 600 (fig 16): Please correct the green arrows color, they are nowhere to be found in the plot.

---

## Referee Comment (RC2) · Anonymous Referee #2 · 21 Nov 2018

**Review on "Sinkholes. Stream channels and base-level fall: a 50-year record of spatio-temporal development of the eastern shore of the Dead Sea" by Watson et al.**

In the present paper Watson et al. present documentation of subtle geomorphological features of the area Ghor-El-Hadita which is located in the east coast of the Dead Sea and suffers severely from infrastructure damages due to the shrinkage of the Dead Sea. These damages include incision of new stream channels and their propagation and steepening, and formation of sinkholes and subsidence areas. Watson and his colleagues have ortho-rectified optical satellite images and aerial photographs of this area in order to describe some of the geomorphological features generated by the coastline retreat and level drop of the Dead Sea in the last 50 years. Beside the morphological features, Watson et al. present also results from measurements of water properties from the sinkholes and the creeks, such as electrical conductivity, a proxy of salinity, and hydrogen and oxygen isotopes, for evaporation degrees.

Although I find this paper as an important documentation, I think that at the present form it is too descriptive with a lack of novel insights and/or understanding of the processes linked to the Dead Sea level fall. Furthermore, the paper presents a variety of sorts of observations with no inter-relations, which give the impression that this paper is a "heap" of arbitrary observations without a purpose. A description of abundant of phenomena related to the Dead Sea level drop in not novel. In the following paragraphs I explain my major concerns, and suggest ways for a significant improvement of the paper, in order to make this paper publishable.

**Morphology of stream channels and alluvial fans**

Here my major concern is that the authors do not present any temporal development in the sinuosity and other channels properties, which is in contradiction to main purpose of the paper. Without such documentation they miss the dynamic processes generated by the DS level fall. The authors describe the present sinuosity of several stream channels, what for? For instance, they show photographs and sketches from four different years, 2000, 2006, 2012, 2017 (Figure 4), and they do not quantify the sinuosity on these dates. I would expect to see temporal variations of the channel sinuosity from these dates, and then to connect it to slope, channel length etc.. They also show the eastward propagation of these channels, which means a migration of the knickpoint of the channel or incision rates. What can they say about? Do they see, through the various dates, shift from meandering to braided channels?

In their variety of geomorphological observations the authors do not show rates (temporal development), beside the migration of the alluvial fan front. I reckon that the papers of Dente et al. (2017) in JGR-ES and Dente et al., in press in ESPL, may provide insights of how to deal with spatiotemporal variations in sinuosity and stream incision under conditions of such a rapid sea level fall.

A surprising observation in this paper is the independency of the sinuosity on the slope (z/L)(Schum, 1993, *Journal of Geology*, who shows that there is a dependency in the slope). They present it in their Figure 6. How did they measure the z/L? is the z/L incremental and for that point they measured the sinuosity? I would think that it is more relevant to put the total z/L of a channel (or segments with major change in slope) and to compare the sinuosity vs. slope among all channels, including CM-8 (revise Figure 6).

**Sinkholes**

1) The authors present here the temporal development of the sinkholes in the area of Ghor-el-Hadita. No doubt an important documentation, I would prefer to see the marks of real sinkholes, i.e., contours along the sinkholes boundaries, as in Abelson et al. (2017, 2018), rather than mere circles. But OK, let's leave that way, but in general, the area of sinkholes is the more credible proxy for sinkholes development rather than their number, as noted by Abelson et al. (2017). The authors' major conclusion is the westward migration of sinkholes activity that follows the retreat of the DS shoreline, similarly described before for the west side of the Dead Sea. I think that the most intriguing observation is the prominent *northward* propagation of the sinkhole activity (see their Fig. 9). I did not see in the paper any notification about this major observation neither any discussion that tries to cope with it.

My suggestion: It is well known from the DS west coast that the sinkhole strips mark the edge of a massive salt layer, the source for the DS sinkholes (see the studies of Ezersky et al. and Abelson et al.) (*Polom et al. [2018] did not find the massive salt layer because their profiles were east of the sinkhole strip, beyond the eastern boundary of the salt layer, and across the strip of the densely populated sinkholes where the salt layer was mostly dissolved. It is pretty obvious that if they would conduct one of their profiles parallel to and west of the sinkhole strip they would observe the massive salt layer -  as usually found in the DS west coast*). It also appears that the sinkhole strip (or the eastern boundary of salt layer) is skewed (in plan view) relative to the shorelines sketched in Fig. 3. So if they will put, lets say the shorelines of 1992, 2000, and 2012 will be enough, on the map of the sinkholes (Fig. 9) the mechanism for the northward sinkhole propagation will pop into our eyes. I mean, the DS shoreline retreat and the skewness of the salt layer boundary relative to the shoreline are causing this northward migration of the sinkhole activity.

2) The authors show various data sets of the sinkholes geometry, e.g., depth vs. diameter (d/r) and sinkhole eccentricity (their Fig. 11). The d/r is shown nicely for different sedimentary environments with reasonable explanation. Still, I do not see the purpose of the eccentricity presentation, and, accordingly, nothing is mentioned in the discussion. I would suggest to put the sinkhole long axis on Rose diagram, to see whether or not there is a preferred orientation. I reckon that there such an orientation. Then to see whether the eccentricity is related to adjacent slope (long axes can be parallel to strike of slope due gravitational stresses), in terms eccentricity versus z/L. Briefly, to purpose is to see whether the exposure of steeper slope may influence the shape of sinkholes.

3) Uvalas – what can the relationship between the uvalas and the sinkholes tell us about the underground cavities? See for instance Atzori et al. (2015, *GRL*).

**Chemistry and salinity of water**

I am not sure that all these sections on the water salinity/chemistry are needed to this paper. There are too many problems with this part, and way to relate to the geomorphological features is not clear.

First, the authors connect the isotopic signatures to salinities. Data Cl concentration is indispensable for such claim. In addition, conductivity measurements for salinity can be tricky. According to Yechieli (2000, *Groundwater*), in the brines around the Dead Sea, conductivity reflects salinity only up to TDS=170 g/l (the DS salinity is ~340g/l). Beyond this value conductivity decreases with salinity increase. So the conductivity measurements in the ponds and springs must be accompanied with salinity measurements. A good and basic measurement for the water chemistry is the Na/Cl ratio. An increase in this ratio tells that the water dissolved salt.

Therefore, the whole part of the water chemistry should be published separately, in other paper. After all this paper tries to show the consequences of the rapid level fall. Bringing all aspects of this fall without new insights on the dynamics of the related processes, loses the advantages of the observations brought here.

**Minor comments**

- Figure 6 that summarizes the geomorphological properties of the channels is very important to deliver major insights arising from this study. Therefore, several improvements are required. First, the dots must much larger, and better to draw different marks for the various channels properties/environments, e.g., meandering, salt, straight, vs. mud , etc.. How did they measure z/L and what is the portion of the sinuosity (in Fig. 6D).
- Figure 11 –Mark A-C on panels. Explain what E means in the lower panel.

---

## Author Comment (AC1) · 24 Apr 2019

We thank the reviewer for the time, effort and consideration put into providing this detailed critique of our manuscript. We address the points made in their review below.

*General Comments*

*The manuscript shows a detailed work on geomorphological processes related to the Dead Sea base-drop for the last 50 years as they express in alluvial incision and sinkholes formation at Ghor al-Haditha, Jordan. The methods include topographic analysis, orthophoto analysis, hydrological isotopes analysis and field observations. The research shows some findings of channel morphology as expected from the local slope, strata and inflow and sinkhole formation in accordance with previous findings along the western Dead Sea shores.*

**C1.1** A major missing component (as admitted by the authors) is an analysis of the hydrological boundary of the fresh-saline water in particular and the underground water levels and composition in general.

**Reply:** We agree in general with this criticism from Reviewer 1, but in the context of the revised manuscript we regard it as a minor issue. The lack of direct constraints on the fresh-saline interface is a limitation of our work, and we retain this statement in the revised version of the manuscript. However, this issue cannot, and we contend that it need not, be addressed directly by the current study. There are no direct subsurface constraints on the position of the fresh-saline interface via boreholes in the study area. While such a direct analysis of the fresh-saline interface would be ideal, we note in the introduction of the revised manuscript that the hypothesis of the fresh-saline interface and its effects on karstification is testable indirectly via its prediction of migration of new sinkhole or uvala development. We present a detailed dataset that enables such a test, and we find that our observations accord well with the hypothesis – better so in extent and consistency than any similar data previously assembled on the western Dead Sea shore. Therefore, we do not accept that this missing aspect of direct analysis of the fresh-saline interface should be a barrier to debate or, indeed, to publication of the revised manuscript at this stage. We argue that it is not required - at this stage - to substantiate the main interpretations and conclusions made in the revised manuscript. As we note in the revised manuscript, this aspect can only be addressed definitively in the future by a drilling program, for which our study provides an improved scientific rationale.

**C1.2** In addition, the authors fail to properly contextualize the results with previous findings along the Dead Sea area. The manuscript and the readers will benefit from detailed comparison with similar results described in the papers already cited in the current manuscript.

**Reply:** in light of this comment from Reviewer 1 and similar comments made by Reviewer 2, we have redefined the focus of the manuscript. The manuscript is now centred on the inter-relationship between subsidence phenomena across several orders of magnitude of scale and the inter-connection between the spatio-temporal evolution of these landforms and the fall in the Dead Sea's hydrological base level. We hope that this re-focussing of our work will help to place the findings in a clearer context and will alleviate any concerns relating to a lack of discussion relating to similar work conducted previously in the Dead Sea region.

**C1.3** Furthermore, the isotopic analysis is incomplete, its results are faintly included in the discussion and cannot support any conclusion regarding salinity. Na/Cl content for example would provide more conclusive information regarding dissolution processes.

**Reply:** After consideration of this criticism from Reviewer 1, we agree that further work is required on this topic. Also, we have decided that this part of the manuscript is surplus to the aims of the manuscript, especially given our decision to focus on the karst geomorphology, and so we have removed this section entirely.

**C1.4** Overall, I find this paper findings to be of much interest showing the Dead Sea base-drop effects are similar on similar environments on either side of the Dead Sea. However, in its current form, I find it is more of a summary of observations and has limited scientific value. I would suggest a major revision and addressing the points below…

**Reply:** The criticism of Reviewer 1, echoed by Reviewer 2, that the original manuscript read as "more of a summary of observations", has spurred us to focus the manuscript on the karst-related geomorphology, on the more generic aspects of linkages between the sinkholes and uvalas, and on their relationship to the Dead Sea base-level drop. We trust that this focussing of the manuscript has enhanced and clarified the scientific value of the study.

*Specific Comments*

**C1.5** Line 31: The response of the surface and subsurface hydrological systems to the base-level drop have been presented previously by e.g. Arakin et al., 2000 env. geol.; Bowman et al., 2007, Geomorphology; Avni et al., 2015, JGR; Shviro et. al., 2017, Geomorphology; I would suggest avoiding using the term "first" here, or explain in detail this research novelty in this context.

**Reply:** our contention in the original manuscript was that we were (to our knowledge) the first to combine the geomporphological study of **both** surface erosional processes (stream channel formation) **and** subsurface development of a karst system, and thus establishing the spatio-temporal links between the two systems in the Dead Sea region. However, in light of the comments from both reviewers, we have undertaken major revisions necessitating the abandonment of this line of argument to focus more upon the subsidence phenomena present at Ghor Al-Haditha.

**C1.6** Line 142: Some error estimations should be provided for the co-registration as done for the DSMs. I'm concerned 9 GCPs are not enough for proper geocoding.

**Reply:** we have now included some tables of root mean square error (RMSE) error estimations in the supplementary material, along with metadata pertaining to the satellite image acquisition.

**C1.7** Line 189: Please add a theoretical line, based on water level drop and slope. I suspect the non-linearity origin is from the non-linearity of the water-level drop rates. As it is described now, one might think it is an abnormal observation, while it might be an expected one. If it does not in agreement with the expected line, a more detail discussion should be added.

**Reply:** We have added this theoretical line to the figure as suggested (Now revised figure 2D). In fact, this addition has proved that the non-linearity of shoreline retreat originates primarily from the non-linearity of the Dead Sea bathymetry rather than the non-linearity of the Dead Sea level drop, as we argued initially.

**C1.8** Lines 291-299: I would suggest putting the sinkhole morphology in context with previous (similar) findings from the western Dead Sea shore. This will strengthen the globality of the findings and put them in proper context rather than highlighting a very local phenomena.

**Reply:** We appreciate this suggestion, and have incorporated the results of Filin et al. (2011) into the revised figure 5B (formerly figure 11B; plot of depth-diameter of sinkholes at Ghor al-Haditha and sites studied by Filin et al. on the western shore).

**C1.9** Line 401: It is not clear why there should be higher evaporation in the salt-edge ponds with respect to mud-edge ponds? They are situated in very close proximity and same environmental conditions. Further water composition analy-sis would be useful for determine if water samples are of evaporative fractionation or mixing of different compounds.  I suspect the difference between the two pond types is mainly due to salt dissolution.  In addition, the isotopic result is not included in the discussion and have little to no support to the conclusions. I would suggest expanding the isotopic and hydrochemistry analysis and to include it in the interpretation. An example of such analysis could be found in e.g. Avni et al., 2016.

**Reply:** As noted in the reply to Comment C1.3 above, we agree with Reviewer 1 that the analysis of water geochemistry could be expanded, but this would necessitate a paper dedicated to that topic. Therefore, and given that it is not critical to our arguments in the revised manuscript, we decided to remove the hydrogeochemistry data from this manuscript.  We aim to return to it in more detail in a future publication.

**C1.10** Line 438: In line 426 it is stated the northern part has steeper bathymetry and here that they are similar.

**Reply:** we have removed this confliction from the manuscript, and indeed the section it was a part of previously in the discussion no longer has a place in the manuscript.

**C1.11** Line 440: Discharge rates are only quoted for the meandering channels and no information is provided for flash floods. I fail to understand how sediment load is related to the morphology. Here you refer the sediment deposits only to support the assumption of the discharge rates. I would suggest obtaining estimations of flash floods discharges to support this assumption. Could the coarser sediments might be originally forming the channel beds and not transported by flesh-floods?

**Reply:** Following Comments C2.2 & C2.3from Reviewer 2, and in light of an extended analysis of the stream channel geomorphology that we have undertaken, we have removed this part of the manuscript and re-focussed our work on the karst-related subsidence phenomena. The above comment is thus immaterial to the revised manuscript. However, in general the nature and concentration of the sediment load is linked with channel morphology and with discharge (see review by Buffington, J. and Montgomery, D. (2013) 'Geomorphic classification of rivers', Treatise on Geomorphology, 9, pp. 730–767. doi: 10.1016/B978-0-12-374739-6.00263-3.). Information about discharge rates of flash floods at Ghor Al-Haditha is non-existent. The coarser sediments we reported were not in place prior to channel incision (thus comprising the channel bed), as they are confined to the channel as unconsolidated bars or braided deposits and no similar deposits are observed in the channel marginal materials (lacustrine marls and evaporites).

**C1.12** Lines 512-513: These findings should also be discussed in context of Baer et al., 2018 (doi: 10.1002/2017JF004594) findings.

**Reply:** We disagree. That paper by Baer et al. (2018) is focussed on a much shorter time interval in the development of individual sinkholes (subsidence precursory to collapse on weeks-months) than can

be resolved in our data. We cannot say much, if anything, about their proposals or model from our data.

**C1.13** Lines 532-536: The depth of the water table in the area and that of the Halit deposits (if present) are required to make this comparison between shallow limestone karst and the Dead Sea Uvalas. Without additional data, the depressions are "widening without deepening" where the base-level fall can be as easily explained by the fact that the karstic layer (Halit) is limited in its thickness as observed on the western side of the Dead Sea (e.g. Ychieli at al., 2016).

**Reply:** we do not say that we have quantitative evidence of the relationship between the evolution of uvalas at GAH and the depth of the water table. We have attempted to clarify this in the revised manuscript (see line 505). Additionally, we have removed all references to the idea of 'widening without deepening', to avoid any confusion or false interpretation. However, we do not feel that Reviewer 1 has understood the point we wished to make, which was that the landforms observed at Ghor Al-Haditha contribute to the understanding of what defines an uvala in a morphometric sense. Indeed, this line of research into the geometric and genetic properties of uvalas and how it relates to proposed mechanisms for their formation is now expanded and forms a central part of the revised manuscript.

**C1.14** Line 541: I fail to see the new insights here. The link between the Uvales formation and sinkhole process is documented in several pervious papers sited in the manuscript.

**Reply:** we disagree with Reviewer 1 in this case. None of the previous studies of uvala formation in evaporite karst, either at the Dead Sea or elsewhere, have systematically studied the morphological links between uvalas and sinkholes in as much spatio-temporal detail or with the same approach (considering links to similar landforms in other karst settings) that we have pursued. Indeed aside from an extremely brief mention in the review by Frumkin (2013), the term 'uvala' does not appear in any of the paper we cited related to the Dead Sea. The new insights should be very clear now in light of the major re-focussing of the manuscript to deal with the geometric and genetic relationships between sinkholes and uvalas.

**C1.15** Line 556: The statement "Evidence . . . is weak" is simply wrong. See for example Avni et al., 2016, figure 6. The seaward shift with time is much more pronounced that in the current paper.

**Reply:** we did not wish to imply that we believe that there is no evidence for this theory in other published works. We only wanted to highlight that there is some disagreement between authors over the significance of that evidence. Charrach (2018), for example, states that he views the evidence for such a migration on the western shore to be weak. This sentence has now been moved to the introduction (line 89 of the revised manuscript), and contextualised accordingly. Regarding the reviewer's last point here, the sinkhole migration observed at Ghor Al-Haditha is greater in spatial magnitude, is observed over a longer timescale, and is overall more consistent in nature than is the case in any study conducted on the western shore (including the work of Avni et al. 2016).

**C1.16** Line 559: I cannot see why this is a stronger evidence than that of e.g. Abelson et al., 2017. Without any information on the fresh-saline interface, it cannot support this theory. Channeling may explain the observations much as well (Arakin et al., 2000) without any evidence of a salt layer and dissolution processes.

**Reply:** we have revised our contention that it is the 'strongest evidence yet', in line with Reviewer 1's concern. We did not wish to place undue emphasis on our results, though we do feel that they provide very convincing evidence that a seaward shift of the fresh-saline interface induced by base-level fall is

a key control on sinkhole development at Ghor Al-Haditha (see lines 530 - 534 in the revised manuscript). Moreover, we do not speculate that the formation and migration of sinkholes at Ghor Al-Haditha is controlled by only dissolutional processes or solely physical erosion in the subsurface. Indeed, our results suggest that a combination of both processes is required to explain the evolution of the sinkhole population at Ghor Al-Haditha, as reported previously by Al-Halbouni et al. (2017). It is difficult to see how channeling can take effect until significant secondary porosity has been created by dissolution. Once a well-connected secondary porosity is developed, then a feedback of further dissolution and channeling, with physical erosion also, can occur.

**C1.17** Line 564: The findings of Polom et al., 2018 of missing slat *(sic)* layer in the fan area may indicate a local area on increased fresh water streaming and accelerated dissolution that removed the salt layer in that area by the time of survey. These results, should be considered with much care for inferring general process related conclusions. The conductivity and mineral contents of the water samples may indicate dissolution processes which is in contrast with Polom et al., findings. A more detailed hydrological analysis may better resolve this issue. The fact that with time, sinkhole distribution is along the whole area, (almost) without gaps, along a very distinct sub parallel line to the shore indicates the possible presence of an underlying salt layer undergoing dissolution processes.

**Reply:** we have removed the section of the manuscript pertaining to the water geochemistry, and therefore we will not comment on the possibility of variations in the ionic and isotopic composition of groundwater across the study area. The study of Polom et al. (2018) does not preclude the presence of salt. It only precludes the presence of a thick (>2 m), continuous salt layer at the time of survey and only in one part of the study area. Our interpretation of the sinkhole distribution and the migration of new sinkhole formation is compatible with either salt concentrated in a single thick layer or salt distributed as many small layers within the marls. We have revised our discussion of this topic in light of this comment and further comments made by Reviewer 2 (see section 5.4; new figure 9).

*Technical comments*

**C1.18** Line 97: "there three" should be "there are three". In general to all figures with topographic data: I would suggest overlaying the color coded elevation over a hillshaded elevation to better express fine detailes.

**Reply:** typo corrected. Many of the figures that previously lacked clarity in a 3D sense due to the absence of a hillshade have now been removed, aside from revised figure 3 where we did not feel it was appropriate to take this step as it would have undermined our efforts to highlight the differencing of the two DSMs. The figures of the elevation of the uvalas have now been modified to show the topography obliquely and thus better highlight the 3D qualities of the data (revised figures 6 and 7).

**C1.19** Line 144: Please add a proper citation to the GDAL library.

**Reply:** done.

**C1.20** Line 484: "is agreement" should be changed to "is in agreement"

**Reply:** done.

**C1.21** Line 600 (fig 16): Please correct the green arrows color, they are nowhere to be found in the plot.

**Reply:** figure removed.

---

## Author Comment (AC2) · 24 Apr 2019

*Response to Anonymous Reviewer #2 comments*

We thank the reviewer for the time, effort and consideration put into providing this detailed critique of our manuscript. We address the points made in their review below.

**General Comments**

*In the present paper Watson et al. present documentation of subtle geomorphological features of the area Ghor-El-Hadita which is located in the east coast of the Dead Sea and suffers severely from infrastructure damages due to the shrinkage of the Dead Sea. These damages include incision of new stream channels and their propagation and steepening, and formation of sinkholes and subsidence areas. Watson and his colleagues have ortho-rectified optical satellite images and aerial photographs of this area in order to describe some of the geomorphological features generated by the coastline retreat and level drop of the Dead Sea in the last 50 years. Beside the morphological features, Watson et al. present also results from measurements of water properties from the sinkholes and the creeks, such as electrical conductivity, a proxy of salinity, and hydrogen and oxygen isotopes, for evaporation degrees.*

**C2.1** Although I find this paper as an important documentation, I think that at the present form it is too descriptive with a lack of novel insights and/or understanding of the processes linked to the Dead Sea level fall. Furthermore, the paper presents a variety of sorts of observations with no inter-relations, which give the impression that this paper is a "heap" of arbitrary observations without a purpose. A description of abundant of phenomena related to the Dead Sea level drop in not novel. In the following paragraphs I explain my major concerns, and suggest ways for a significant improvement of the paper, in order to make this paper publishable.

**Reply:** in light of this same concern outlined by both reviewers, we have redefined the focus of the manuscript to highlight and expand upon our discoveries made regarding the process of uvala formation and the morphological relationship between uvalas and sinkholes. We incorporate this into a broader understanding of subsidence phenomena and processes linked to evaporite karstification, as observed at Ghor Al-Haditha on the Dead Sea shore. We also present numerous novel insights into the spatio-temporal aspects of these processes and their relationship to the declining hydrological base level at the Dead Sea. Implications for similar processes (albeit over much longer timescales) in limestone karst are also discussed, thus broadening the scope of the original manuscript. Much of the 'abundant description' that was present in the previous version of the manuscript has now been omitted to provide a sharper focus to the results and to present a more coherent story to accompany them.

*Specific Comments*

**C2.2** …the authors do not present any temporal development in the sinuosity and other channels properties, which is in contradiction to main purpose of the paper. Without such documentation they miss the dynamic processes generated by the DS level fall. The authors describe the present sinuosity of several stream channels, what for? For instance, they show photographs and sketches from four different years, 2000, 2006, 2012, 2017 (Figure 4), and they do not quantify the sinuosity on these dates. I would expect to see temporal variations of the channel sinuosity from these dates, and then to connect it to slope, channel length etc. They also show the eastward propagation of these channels, which means a migration of the knickpoint of the channel or incision rates. What can they say about? Do they see, through the various dates, shift from meandering to braided channels? In their variety of

geomorphological observations the authors do not show rates (temporal development), beside the migration of the alluvial fan front. I reckon that the papers of Dente et al. (2017) in JGR-ES and Dente et al., in press in ESPL, may provide insights of how to deal with spatiotemporal variations in sinuosity and stream incision under conditions of such a rapid sea level fall.

**Reply:** we agree with Reviewer 2 that such a temporal analysis as provided by Dente et al (2017) and Dente et al (2018) is a good way to analyse the spatio-temporal evolution of channel morphological properties. Having now performed the analysis as suggested by Reviewer 2, and some additional analysis, we have realised that a substantial number of new figures is required to illustrate the stream channel evolution and thus that it requires its own treatment in a dedicated manuscript. We have therefore removed all quantitative spatio-temporal analysis of the stream channels to better focus the revised manuscript.

**C2.3** A surprising observation in this paper is the independency of the sinuosity on the slope (z/L) (Schumm, 1993, Journal of Geology, who shows that there is a dependency in the slope). They present it in their Figure 6. How did they measure the z/L? is the z/L incremental and for that point they measured the sinuosity? I would think that it is more relevant to put the total z/L of a channel (or segments with major change in slope) and to compare the sinuosity vs. slope among all channels, including CM-8 (revise Figure 6).

**Reply:** in line with the suggestions given by Reviewer 2, we made a number of revisions to this aspect of the analysis as presented in the original manuscript. One of these was to divide each channel into 10 reaches of arbitrary length to analyse the spatio-temporal evolution of stream channel W/D, Z/L and sinuosity in a more relatable fashion. Much of the analysis was also re-done to check the accuracy of the claims made previously. The result is that we have generated several additional figures to document the variation in stream channel width and sinuosity through time and to explain how the patterns we see may relate to factors such as substrate type, 'valley' slope, discharge and base level fall. Essentially, a proper treatment of the stream channel requires its own manuscript, which we suggest we return to in the future. Therefore, we decided to refocus the paper to deal solely with the subsidence phenomena observed at Ghor Al-Haditha and removed figure 6 from the manuscript entirely.

**C2.4** The authors present here the temporal development of the sinkholes in the area of Ghor-elHadita. No doubt an important documentation, I would prefer to see the marks of real sinkholes, i.e., contours along the sinkholes boundaries, as in Abelson et al. (2017, 2018), rather than mere circles. But OK, let's leave that way, but in general, the area of sinkholes is the more credible proxy for sinkholes development rather than their number, as noted by Abelson et al. (2017).

**Reply:** we agree with Reviewer 2 regarding this point. However, we fell that the spatio-temporal analysis of sinkhole migration as presented in figure 4 (formerly figure 9) is itself sufficient to support the hypothesis of a migration of the interface between fresh groundwater and hypersaline Dead Sea water. In light of our re-focus away from the local aspects of this study to give more weight to our discoveries regarding inter-relations between sinkhole and uvala morphology and formation, we have removed references to growth of sinkhole population from the manuscript (removed former figure 10). We hope to return to this aspect in a later publication.

**C2.5** The authors' major conclusion is the westward migration of sinkholes activity that follows the retreat of the DS shoreline, similarly described before for the west side of the Dead Sea. I think that the most intriguing observation is the prominent northward propagation of the sinkhole activity (see their Fig. 9). I did not see in the paper any notification about this major observation neither any

discussion that tries to cope with it. My suggestion: It is well known from the DS west coast that the sinkhole strips mark the edge of a massive salt layer, the source for the DS sinkholes (see the studies of Ezersky et al. and Abelson et al.) (*Polom et al. [2018] did not find the massive salt layer because their profiles were east of the sinkhole strip, beyond the eastern boundary of the salt layer, and across the strip of the densely populated sinkholes where the salt layer was mostly dissolved. It is pretty obvious that if they would conduct one of their profiles parallel to and west of the sinkhole strip they would observe the massive salt layer - as usually found in the DS west coast*). It also appears that the sinkhole strip (or the eastern boundary of salt layer) is skewed (in plan view) relative to the shorelines sketched in Fig. 3. So if they will put, lets say the shorelines of 1992, 2000, and 2012 will be enough, on the map of the sinkholes (Fig. 9) the mechanism for the northward sinkhole propagation will pop into our eyes. I mean, the DS shoreline retreat and the skewness of the salt layer boundary relative to the shoreline are causing this northward migration of the sinkhole activity.

**Reply:** we are in full agreement with Reviewer 2 regarding this proposed explanation for the northward migration of the sinkholes with time as shown in figure 9 (now figure 4). Therefore, and given the relaxed space requirements arising from our focussing on the sinkhole and uvala development, we have added a discussion of this theory to the revised manuscript (section 5.4; new figure 9).

**C2.6** The authors show various data sets of the sinkholes geometry, e.g., depth vs. diameter (d/r) and sinkhole eccentricity (their Fig. 11). The d/r is shown nicely for different sedimentary environments with reasonable explanation. Still, I do not see the purpose of the eccentricity presentation, and, accordingly, nothing is mentioned in the discussion. I would suggest to put the sinkhole long axis on Rose diagram, to see whether or not there is a preferred orientation. I reckon that there such an orientation. Then to see whether the eccentricity is related to adjacent slope (long axes can be parallel to strike of slope due gravitational stresses), in terms eccentricity versus z/L. Briefly, to purpose is to see whether the exposure of steeper slope may influence the shape of sinkholes.

**Reply:** after examining the data as suggested by Reviewer 2, we found that there is indeed a preferred orientation for long and short axes of sinkholes, but that this is not parallel to the strike of slope. Instead, it is broadly parallel to the aspect of the slope - i.e. perpendicular to the strike of slope. Please see revised figure 5D (newly added to what was formerly figure 11) and revised section 4.3 for a summary of these findings.

**C2.7** Uvalas – what can the relationship between the uvalas and the sinkholes tell us about the underground cavities? See for instance Atzori et al. (2015, GRL).

**Reply:** we do not feel that our results in themselves provide us with much basis to discuss this relationship, as we cannot independently identify sub-surface cavities prior to a collapse. Based on the DEM models presented for subsidence around multiple void spaces by Al-Halbouni et al. (2019, in review), the uvalas seem to represent an integrated subsidence response to the development of many smaller-scale cavities in the subsurface as opposed to a single $10^2$ scale cavity. There is no clear preference for new sinkholes to form at the margins of uvalas, as might be expected from the model of Atzori et al. We suspect that the reality is more complex than their model.

**C2.8** I am not sure that all these sections on the water salinity/chemistry are needed to this paper. There are too many problems with this part, and way to relate to the geomorphological features is not clear. First, the authors connect the isotopic signatures to salinities. Data Cl concentration is indispensable for such claim. In addition, conductivity measurements for salinity can be tricky. According to Yechieli (2000, Groundwater), in the brines around the Dead Sea, conductivity reflects

salinity only up to TDS=170 g/l (the DS salinity is ~340g/l). Beyond this value conductivity decreases with salinity increase. So the conductivity measurements in the ponds and springs must be accompanied with salinity measurements. A good and basic measurement for the water chemistry is the Na/Cl ratio. An increase in this ratio tells that the water dissolved salt. Therefore, the whole part of the water chemistry should be published separately, in other paper. After all this paper tries to show the consequences of the rapid level fall. Bringing all aspects of this fall without new insights on the dynamics of the related processes, loses the advantages of the observations brought here.

**Reply:** after some consideration and in light of the comments made by Reviewer 1 regarding these sections, we have decided that this part of the manuscript is indeed surplus to the aims of the manuscript and does not provide any additional insight to the narrative presented. Therefore, we have removed this section entirely from the manuscript.

*Technical comments*

**C2.9** Figure 6 that summarizes the geomorphological properties of the channels is very important to deliver major insights arising from this study. Therefore, several improvements are required. First, the dots must much larger, and better to draw different marks for the various channels properties/environments, e.g., meandering, salt, straight, vs. mud , etc.. How did they measure z/L and what is the portion of the sinuosity (in Fig. 6D).

**Reply:** as stated in the reply to reviewer comments **C2.2 and C2.3,** we decided to refocus the paper to deal solely with the subsidence phenomena observed at Ghor Al-Haditha and so we removed figure 6 from the manuscript entirely.

**C2.10** Figure 11 –Mark A-C on panels. Explain what E means in the lower panel.

**Reply:** done; see modified figure 11 (now figure 5).

---

## Referee Report (RR1)

[referee-annotated manuscript omitted]

---

## Author Response (AR2)

Dear Drs Krawczyk and Agnon,

I write to you in accompaniment of the submission of a revised version of our manuscript in Solid Earth Discussions, Watson et al., 2019 ('*Sinkholes and uvalas in evaporite karst: spatio-temporal development with links to base-level fall on the eastern shore of the Dead Sea*'). In light of comments from reviewer 3 (Jo De Waele) and Dr Agnon we have made changes to the manuscript, as highlighted below.

The revised manuscript has been updated in several respects. Firstly, we have attempted to clarify in the Introduction the novelties of our research in the context of previous work undertaken in the study area. We believe that the present manuscript offers results that are much more extensive in space and time: our results yield the most detailed insights to date into the spatio-temporal development of sinkholes and uvalas in evaporite karst settings, and they provide the clearest yet illustration of the consequences of base-level fall on that development.

It was also suggested by reviewer 3 that a more thorough characterisation of the Quaternary sediments in which the sinkholes and uvalas are formed was also required. In order to address this, we have modified Figure 1c to indicate the extents of the different materials on the surface, included a further supplementary figure (Figure S1) showing field impressions of the different sedimentary materials, and expanded the descriptions of the deposits (lines 155–177 of the revised manuscript).

To highlight the insights/novelties of the work, to deal clearly with the reviewer's comments, and to improve the logical flow of the manuscript, we have rewritten and restructured the Introduction and some of Discussion sections of the manuscript. We have added a new figure (Figure 10 of the revised manuscript) to highlight the insights the manuscript gives into the processes governing uvala formation in the study area (i.e. to provide a more visual 'take-home message'). We emphasise that these revisions are essentially editorial in nature; they have not resulted in any changes either to the overall direction or to the main scientific findings of the work.

In accompaniment to the revised manuscript, we also provide a point by point response to the comments of reviewer 3, whom we wish to thank for his detailed and constructive review.

I can confirm that we have no conflicts of interest and that we have no related work submitted or in press anywhere else. The co-authors of the manuscript, as listed on its first page, have all consented to this revised submission to Solid Earth. Please do not hesitate to contact me should you require further information.

Yours sincerely,

Robert A. Watson

We thank the reviewer for the time, effort and consideration put into providing this detailed critique of our manuscript. We address the points made in his review below.

*General Comments*

**C3.1** You published two papers on the same research area in Solid Earth (you cite them). What makes this new paper different and novel enough from the other two and worth publishing in Solid Earth? This aspect makes this paper seem less novel than it really is.

**Reply:** This new manuscript is distinct in scope and subject to the works published previously in Solid Earth and other journals. It expands upon the work presented by Al-Halbouni et al., 2017 (doi: *https://doi.org/10.1016/j.geomorph.2017.02.006*) as it presents an expanded set of data both spatially, to cover the entire study area, and temporally, as it documents landscape evolution at a much higher temporal resolution and over a longer time period. It also provides an empirical, 'field laboratory' set of findings that contextualise the findings of the numerical modelling studies presented by Al-Halbouni et al. (2018 and 2019, both in Solid Earth) and the geophysical study presented by Polom et al. (2018, also in Solid Earth). In the Introduction section of the revised manuscript (lines 86–91 in the revised manuscript), we have now clarified how our work expands on previous work. Additionally, in order to better highlight the most novel findings of the work, we have slightly restructured the manuscript by placing discussion section 5.4 earlier (in the revised manuscript it is 5.2) and by adding a new figure (Figure 10 in the revised manuscript) which aims to clarify the main new generic findings of this manuscript.

**C3.2** The paper is well prepared, but I believe there is a lack of more detailed geological information on the types of rocks involved. A general "alluvium", marls, clays etc. might not be sufficient to give the reader a good idea of which geological formations we are talking. I would have expected a detailed sedimentological, mineralogical and petrographical description of the sediments that are related to the sinkhole formation. I do not believe there is no information of a single drill hole, or some outcrops in the sinkholes themselves, that would allow to describe the geological units subdued to sinkhole formation in much more detail. How can these formations be distinguished? Any geochemical-mineralogical data on these formations (% of clay, sand, calcite, gypsum, halite...). I believe this is fundamental information to know how much dissolution can be responsible for void formation, and thus collapse.

**Reply:** We have attempted to expand the information presented on the geological nature of the Lisan and Ze'elim formations by re-drawing the geological map presented in Figure 1c to show the spatial extents of the 'Alluvium', 'Mudflats', and 'Salt-flats' more precisely. Additionally, we present in the revised Supplementary Material a new figure, Figure S1, which gives field impressions of these surface deposits and should help readers to better understand the nature of the Lisan formation. While noting some of the limitations regarding the stratigraphic constraints at the study site, we have also included more details on the sedimentological, mineralogical and petrographical nature of the sediments with reference to existing literature (lines 155–177 in the revised manuscript) that characterise these sediments on both shores of the Dead Sea.

**C3.3** The subdivision into mud, salt and alluvium sinkholes needs to be more described in detail. I suspect there is a gradual change from mud dominated to salt dominated and alluvial sinkholes. This subdivision is somewhat arbitrary (and can bias the analysis) and would need a clear description of

the boundaries between these three classes of sinkholes. This comment is also related to the previous comment. How much % in salt would be needed to call a sinkhole a salt one and not a mud one??? This division has also an impact upon your conclusions and discussions. I suggest to add a figure with the typical examples of these types of sinkholes... and make clear on what your division in three classes is based.

**Reply:** We agree that the division of sinkholes by surface cover material is a somewhat arbitrary way to classify them and that the actual material compositions cannot easily be rigidly defined. The reviewer is correct in his suspicion of a gradual change between materials. However, we feel that the classification is a useful way to contextualise the results in light of previous studies (Al-Halbouni et al., 2017, 2018; Filin et al., 2011). To help the reader understand the system of classification, we have updated Figure 1c to show the spatial extents of the 'Alluvium', 'Mud', and 'Salt' deposits at the surface. Additionally, we present in the revised Supplementary Material a new figure, Figure S1, which gives field impressions of these surface deposits to emphasise the differences between the materials. We have clarified our methodology for classifying the sinkholes and acknowledged the limitations raised by the reviewer, in part with reference to Figures 2 and 3 of (Al-Halbouni et al., 2018) which shows material-linked end-members and gradations of sinkhole morphology at the study site (lines 208–218 in the revised manuscript).

**C3.4** The comparison of your sinkholes and uvalas with those in limestone (and even gypsum) is somewhat forced. The processes at play in both situations are different (although having dissolution in common). Limestones are hard, and erosion plays a minor role, your sediments are easily erodible. Also times of formation are extremely different. I do not really like your effort of comparison. I would stick to the detailed description of what happened in your area, and focus on the processes at play, and describe the morphology of your sinkholes and uvalas (admitting your depressions can be defined as uvalas!?). In your situation there is a lot of piping involved, I believe. I am not convinced you can really talk about "conduits"...

**Reply:** The purpose of this comparison is to place our results in a broader, more global context – as befits an international journal. Given that this comparison represents about 7 % of the entire main text, we feel that the revised manuscript strikes a good balance between the local and global aspects of the research. Our comparison is clearly acknowledged in the text to be an opening of a discussion regarding the similarities and differences between limestone karst uvalas and evaporite-karst uvalas, rather than any definitive analysis. Our hope that this debate may be continued and progressed in a scientifically rigorous fashion through the assembly and analysis of more extensive, global data sets. We have attempted to clarify our comparisons to limestone karst (section 5.3 in the revised manuscript) and other evaporite karst areas (lines 496–506 in the revised manuscript). In this we also make a clearer case for why our large-scale depression can be regarded as uvalas. We agree that piping may be more of a factor in our study area than may be the case in limestone areas – this we have clarified in the revised text. Regarding the reviewer's doubts concerning the existence of conduits, we draw his attention to Section 4.5, where the evidence for conduits is set out. To help visualise the specific nature of the processes governing sinkhole and uvala formation in our study area, we have added a new figure (Figure 10 in the revised manuscript).

*Specific Comments*

Reviewer 3 provided an extremely thorough commentary of minor revisions (spelling errors, missing references, etc.) to the manuscript, for which we are very grateful. These we have corrected in line with the manuscript revision as a whole.